# Adaptive Constrained Optimization for Neural Vehicle Routing

## Abstract

Neural solvers have shown remarkable success in tackling Vehicle Routing Problems (VRPs). However, their application to scenarios with complex real-world constraints is still at an early stage. Recent works successfully employ variants of the Lagrange multiplier method to handle such constraints, but their limitation lies in the use of a uniform multiplier across all problem instances, overlooking the fact that the difficulty of satisfying constraints varies significantly across instances. To address this limitation, we propose an instance-level adaptive constrained optimization framework that reformulates the Lagrangian dual problem by assigning each instance its own multiplier. To efficiently optimize this new problem, we design a multiplier-conditioned policy that solves instances with a controllable level of constraint awareness, which effectively decouples policy optimization from the optimization of multipliers. By leveraging this conditioned policy, we customize the optimization of multipliers for each test instance by adapting to its particular constraint violations. Experimental results on the Travelling Salesman Problem with Time Window (TSPTW), and TSP with Draft Limit (TSPDL) show that our method exhibits advantages compared to the strong solver LKH3 and significantly outperforms state-of-the-art neural methods. Our code is available at https://anonymous.4open.science/r/ICO-E52F.

## 1 Introduction

The Vehicle Routing Problem (VRP) is a classic kind of NP-hard combinatorial optimization problem with broad real-world applications in manufacturing [62], transportation [57], and logistics [39]. VRP solvers in the Operational Research (OR) community, which are typically based on heuristic search [29] and integer programming [4], have achieved remarkable success in the past but are often limited by high computational overheads. To address this, neural networks have been leveraged to develop efficient, data-driven heuristics for solving VRPs [64, 35, 40, 49, 37, 33, 11, 69, 45], demonstrating faster solving speeds and competitive solution quality against strong OR solvers. A prominent approach among these neural solvers is utilizing reinforcement learning-based policies to sequentially construct solutions [5], which has shown effectiveness on canonical problems like TSP and Capacitated VRP (CVRP) [41, 19, 47].

Real-world applications of VRP, however, often involve constraints that are more complex than those in the canonical problems. For example, in many business scenarios such as public transportation [12, 56] and dial-a-ride systems [16], the arrival time of vehicle must fall into a customer-requested time window, known as the time window constraint. This constraint significantly restricts the feasible region such that even finding a feasible solution is proved to be NP-complete [55], which can pose great challenges to most existing solvers. Other examples of complex constraints in VRPs include the global priority rule in disaster relief [52] and the draft limits in maritime transportation [26]. To

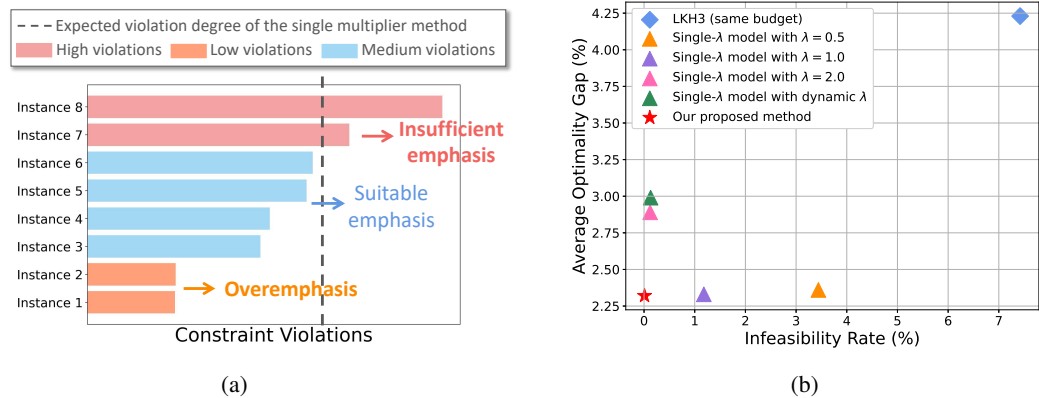

Figure 1: (a) Illustration of the drawback inherent in single-multiplier ($\lambda$) methods. Constraint violations of different instances are plotted. The single-$\lambda$ methods tend to overemphasize (insufficiently emphasize) constraints on some instances with relatively low (high) constraint violations. (b) Performance comparison of LKH3, single-$\lambda$ models and our proposed instance-level adaptive method, on TSPDL with 50 nodes.

handle these hard constraints, classical OR solvers often employ techniques like penalty functions to incorporate constraint violations into the objective function. In the strong solver LKH3 [30], the penalty function is prioritized over the original distance cost, highlighting its emphasis on handling constraints. However, as shown in pervious works [9] and our experiments (see Table 1), the feasibility rate obtained by the traditional solvers is still unsatisfactory when runtime budgets are limited.

Neural solvers have achieved remarkable performance on various VRPs, even surpassing LKH3 on large-scale problems [48] and specific problem variants [72]. However, the research of their extension to VRPs with complex constraints is still at an early stage. To better handle complex constraints, existing studies have refined neural methods from several perspectives, including constraint-aware feature design [15], improvement in network architecture [21], modifications to the objective function [71, 14, 60], and development of novel masking mechanisms [9]. For instance, Chen et al. [15] introduced a multi-step look-ahead strategy, integrating the future time window information to enhance constraint-related features. Similarly, Bi et al. [9] designed a look-ahead-based mask mechanism to proactively exclude actions that would violate constraints in future steps. From the perspective of constrained optimization, Tang et al. [60] adopted the Lagrange multiplier method to explicitly optimize constraint violations together with the route distance. Notably, the most recent Lagrange multiplier-based implementation [9] has achieved state-of-the-art performance on common benchmarks, regarded as a general and effective solution for complex VRPs. However, these Lagrangian-based methods directly extend the canonical formulation to the optimization of neural solvers by employing a uniform multiplier across all problem instances, thereby neglecting the disparity in constraint violations among instances, as illustrated in Figure 1a. This limitation can significantly hinder the adaptability of neural models, resulting in suboptimal performance. More related works about neural solvers and constrained optimization are introduced in Appendix C.

To address this issue, we introduce a new formulation of the Lagrangian dual problem that assigns each instance a specific multiplier, enabling adaptive constrained optimization at the instance level. Compared to the methods that rely on a single multiplier, this instance-specific formulation offers greater flexibility by optimizing the trade-off between solution quality and constraint satisfaction for each individual instance. However, directly optimizing instance-specific multipliers for millions of training instances poses significant computational challenges. To address this issue, we develop a multiplier-conditioned policy that decouples the optimization of the policy from that of the multipliers, effectively reformulating the dual problem into two separate subproblems. By leveraging this conditioned policy, the outer subproblem of optimizing multipliers can be efficiently solved independently during the inference stage.

We conduct experiments on two challenging constrained VRPs: Travelling Salesman Problems with Time Window (TSPTW), and TSP with Draft Limit (TSPDL). Notably, these two problems pose greater challenges in satisfying constraints compared to CVRPTW and CVRPDL, as the

constraint violations of the latter can be addressed more easily by assigning additional vehicles to the violated nodes. The experimental results demonstrate that our adaptive optimization approach significantly surpasses the state-of-the-art neural method [9] that relies on a single multiplier. For instance, Figure 1b compares the optimality gap and infeasibility rate on TSPDL50 (TSPDL with 50 nodes), where our proposed method has clear advantages. Moveover, compared to the strong solver LKH3 under the same runtime budget, our neural method reduces the infeasibility rate by $95.56\% - 1.33\% = 94.23\%$ on TSPTW100 and $7.02\% - 0.91\% = 6.11\%$ on TSPDL100, while achieving competitive optimality gap. These results highlight neural methods as a promising alternative to OR solvers for addressing constrained VRPs.

## 2 Background

### 2.1 Constrained VRPs

The objective of VRPs [18] is to determine a tour that minimizes the total travel distance while visiting all the customer nodes. Formally, a VRP instance is defined on a graph $G = (V, E)$, where $V$ represents the set of all customer nodes along with a depot node, and $E$ denotes the set of directed edges between each pair of nodes (i.e., the graph is fully connected). The vehicles are required to start and end their tours at the depot node. In this paper, we focus on two types of challenging constraints: Time window constraint and draft limit constraint.

**Time window.** The time window constraint nartually arises in many business scenarios that require flexible time scheduling [61]. In this context, each node is accosiated with a time window $[l_i, u_i]$ that defines the earlist time $l_i$ and the latest time $u_i$ of visiting that node. The constraint ensures that the arrival time at each node does not exceed the end of its designated time window. If the arrival time $t_i$ is earlier than the start time (i.e., $t_i < l_i$), the vehicle must wait until the time window starts. Formally, a TSPTW instance $I$ is expressed as:

$$\min_{\tau} f_I(\tau) = \sum_{(n,v) \in \tau} d_I(n, v), \qquad \text{s.t.} \quad g_I(\tau) = \sum_{i=0}^{n-1} \max\{t_i - u_i, 0\} \leq 0,$$

where $\tau$ denotes the tour, and $d_I(n, v)$ is the distance between nodes $n$ and $v$. The goal is to find a tour $\tau$ that minimizes the total distance $f_I(\tau)$ while satisfying the time window constraint $g_I(\tau) \leq 0$.

**Draft limit.** The draft limit in ports is an important factor that influences the routing actions in maritime transportation [26]. The draft of a ship is the distance between the waterline and the bottom of the ship, affected by the cumulative load. The draft limits in ports are designed to avoid overloaded ships entering these ports. In this context, each node represents a port with a maximum draft $m_i$ and a non-negative demand $\delta_i$. The constraint requires that the cumulative load, $c_i = \sum_{j=1}^{i-1} \delta_{\tau_j}$, over the last $i - 1$ steps must not exceed the maximum draft $m_i$ of the $i$-th visited port. Formally, this can be expressed as $g_I(\tau) = \sum_{i=0}^{n-1} \max\{c_i - m_i, 0\} \leq 0$.

### 2.2 Lagrange Multiplier Method

To solve constrained VRPs, the constraint violation can be integrated into the objective function through the formulation of the Lagrangian dual problem [7]:

$$\max_{\lambda \geq 0} \min_{\tau} [f_I(\tau) + \lambda \cdot g_I(\tau)],$$

where $\lambda$ is a non-negative dual variable (i.e., multiplier), quantifing the impact of a constraint on the objective function. The Lagrangian dual problem can be optimized by alternatively updating the primal and dual variables. This involves solving the primal problem for a fixed dual variable, which can be addressed using a classical VRP solver, followed by updating the dual variable based on the observed constraint violations [38]. The update of the dual variable is often realized using subgradient ascent as:

$$\lambda \leftarrow \lambda + \alpha \cdot g_I(\tau),$$

where $\alpha$ is the learning rate. Through the iterative adjustment, the dual variable is continuously refined according to the current level of constraint violation, enabling a better balance between solution

quality and constraint satisfaction. More iterative update methods for the dual variable include quadratic method [31] and proportional-integral-derivative control [58]. Compared to traditional penalty-based approaches, the Lagrange multiplier method avoids reliance on fixed penalty parameters and has the potential to yield optimal solutions if the strong duality holds [10]. However, the Lagrange multiplier method is designed to optimize an individual problem instance. A gap arises nartually when it is applied to the training process involving a large number of instances.

## 2.3 Lagrange Multiplier-based Training Methods for Neural Vehicle Routing

When reinforcement learning (RL) is employed to train neural networks for constructing solutions to VRPs [5], the expected return of the RL policy $\pi_\theta$ on a given instance $I$ is defined as $\mathcal{J}(\pi_\theta, I) = \mathbb{E}_{\tau \sim \pi_\theta(\cdot|I)}[-f_I(\tau)]$, and the expected constraint violation is given by $\mathcal{J}_C(\pi_\theta, I) = \mathbb{E}_{\tau \sim \pi_\theta(\cdot|I)}[-g_I(\tau)]$. Using these definitions, the Lagrangian dual problem of policy optimization is formulated as

$$\min_{\lambda \geq 0} \max_\theta \mathbb{E}_{I \sim \mathcal{D}}[\mathcal{J}(\pi_\theta, I) + \lambda \cdot \mathcal{J}_C(\pi_\theta, I)].$$

Unlike typical constrained RL [1, 68, 28], where the focus is on solving a specific instance, the trained policy in this framework is designed to generalize to unseen instances from the same problem class. To achieve this, the training objective involves maximizing the expected performance over a distribution $\mathcal{D}$ of instances. In practice, the training process is conducted on a dataset $D_I$ that contains a large number of synthetic problem instances.

To optimize this (or a similar) dual problem, Tang et al. [60] proposed an approach that alternatively updates the policy $\pi_\theta$ and the multiplier $\lambda$. Specifically, the policy $\pi_\theta$ is optimized by policy gradient algorithms such as REINFORCE [67], while the multiplier $\lambda$ is optimized by subgradient ascent. More recently, Bi et al. [9] chose to fix the value of $\lambda$ as a pre-defined constant for efficiency and scalability. In our experiments (see Table 1), we observe that dynamically updating the policy-level single $\lambda$ is inferior to the fixed $\lambda$ setting in most cases.

**Limitations of Lagrangian-based training.** The Lagrange multiplier method was originally designed for optimizing a single problem instance. However, existing approaches directly extend this method to the training of neural solvers and ties multipliers to the RL policy, forming a *policy-level* dual approach, where $\lambda$ updates with policy changes but remains invariant across instances. This simple adaptation overlooks the fact that different instances can exhibit significantly varying levels of constraint violations, as demonstrated in Figure 1a, thereby resulting in suboptimal performance.

## 3 Method

To address the aforementioned limitations, we propose an Instance-level adaptive Constrained Optimization (ICO) method. In this section, we first provide an overview of the proposed ICO approach, followed by a detailed description of its training process and network architecture.

### 3.1 Instance-level Adaptive Constrained Optimization

We leverage instance-specific multipliers to effectively handle the varying degrees of constraint violations across instances, which can enable a more flexible trade-off between optimizing the objective and satisfying the constraints. Formally, the new dual problem is formulated as

$$\min_{\{\lambda_i\}_{i=1}^N} \max_\theta \sum_{i=1}^N [\mathcal{J}(\pi_\theta, I_i) + \lambda_i \cdot \mathcal{J}_C(\pi_\theta, I_i)], \tag{1}$$

where $N$ is the number of training instances and $\lambda_i$ is the dual variable specific to instance $I_i$. This dual formulation has the potential to simultaneously improve solution quality and enhance constraint satisfaction, provided that both the primal and dual variables are effectively optimized. However, it is extermely challenging and computationally expensive to optimize the instance-specific dual variables for **millions of training instances**. In the common training method of neural solvers [41], more than one hundred million training instances are generated on the fly, and each instance is only used once during training without additional iterations to refine its corresponding multiplier. This training process necessitates an efficient and scalable approach to adaptively manage instance-specific

multipliers. Therefore, we discard the expensive alternating update method and decouple the original bi-level optimization problem into two separate subproblems: Solve the inner subproblem of Eq. (1) as phase 1 and solve the outer subproblem based on the inner results as phase 2.

**Phase 1: Solve the inner subproblem.** In the first phase, we solve the inner maximization problem separately while considering varying values of $\lambda$, aiming to obtain a manifold of policies capable of solving instances with continuously varying levels of constraint awareness. To achieve this, we propose training a $\lambda$-conditioned policy $\pi_\theta(\cdot|\lambda)$ that takes $\lambda$ as input and performs as trained using the specified $\lambda$, i.e.,

$$\pi_\theta(\cdot|\lambda) \approx \arg\max_\pi \sum_{i=1}^N [\mathcal{J}(\pi, I_i) + \lambda \cdot \mathcal{J}_C(\pi, I_i)],$$

where the right side represents the optimal policy corresponding to the given $\lambda$. With this condition mechanism, the constraint sensitivity of the policy can be seamlessly controlled by adjusting the input value of $\lambda$, without requiring any modification to the network parameters. This can effectively decouple the policy optimization process from the optimization of the multipliers, thereby enhancing scalability of the Lagrangian-based training method. The detailed training algorithm and network architecture for the $\lambda$-conditioned policy are provided in Section 3.2.

**Phase 2: Solve the outer subproblem.** The second phase is performed during the inference stage, where instance-specific $\lambda$ values are optimized based on the feedback provided by the trained $\lambda$-conditioned policy. For each new instance, we iteratively update $\lambda$ by subgradient ascent to minimize its specific constraint violations, thereby adjusting the policy to achieve an appropriate trade-off. This process alternates between sampling a solution using the policy $\pi_\theta(\cdot|\lambda)$ and updating $\lambda$ based on the observed constraint violations. Formally, the process is described as follows:

$$\tau_{t-1} \sim \pi_\theta(\cdot|\lambda_{t-1}, I), \quad \lambda_t = \lambda_{t-1} + \alpha \cdot g_I(\tau_{t-1}),$$

where $t$ denotes the iteration timestep, and $g_I(\tau_{t-1})$ is the constraint violation of the sampled solution. Note that we initialize all $\lambda$ values using an identical $\lambda_0$. Furthermore, we also explore to utilize Proportional-Integral-Derivative (PID) control to adjust the $\lambda$-value as proposed by Stooke et al. [58], detailed in Appendix F.2.

## 3.2 Multiplier-Conditioned Policy

The $\lambda$-conditioned policy serves as a key component in optimizing the decoupled dual problem. We design a two-stage training algorithm for the $\lambda$-conditioned policy, consisting of a pre-training stage for efficient convergence and a fine-tuning stage to achieve a precise alignment between $\lambda$ values and instance hardness, which is schematically illustrated in Figure 2. Detailed description of the two training stages is as follows.

**Pre-training stage.** The pre-training stage is conducted on randomly sampled $\lambda$ values, which is computationally efficient and can effectively enable the generalization ability across varying $\lambda$ conditions. The training objective can be expressed as

$$\max_\theta \mathbb{E}_{I \sim \mathcal{D}} \mathbb{E}_{\lambda \sim \mathcal{D}_\lambda} [\mathcal{J}(\pi_\theta(\cdot|\lambda), I) + \lambda \cdot \mathcal{J}_C(\pi_\theta(\cdot|\lambda), I)].$$

Specifically, we randomly sample $\lambda_i$ from a pre-defined distribution $\mathcal{D}_\lambda$ for each training instance $I_i$, constituting a pair sample $(\lambda_i, I_i)$. The reward function of the instance $I_i$ is reweighted by its own multiplier $\lambda_i$. Following the shared baseline method [41], we sample multiple solutions $\{\tau^j\}_{j=1}^P$ for each $(\lambda_i, I_i)$ pair and estimate the baseline by the average reward of these solutions. Then, we compute the policy gradient $\nabla_\theta J(\theta)$ using the REINFORCE [67] algorithm as

$$R^j = -(f_{I_i}(\tau^j) + \lambda_i \cdot (g_{I_i}(\tau^j) + c_{I_i}(\tau^j))), \forall j \in [P],$$

$$\nabla_\theta J(\theta) = \frac{1}{P} \sum_{j=1}^P (R^j - \frac{1}{P} \sum_{k=1}^P R^k) \log \pi_\theta(\tau^j|\lambda_i, I_i),$$

where $[P]$ denotes the set $\{1, ..., P\}$, and $c_{I_i}(\tau^j)$ is the number of timeout nodes, which we use as a heuristic penalty reward, following the reward design of [9]. The factor $R^j - \frac{1}{P} \sum_{k=1}^P R^k$ represents

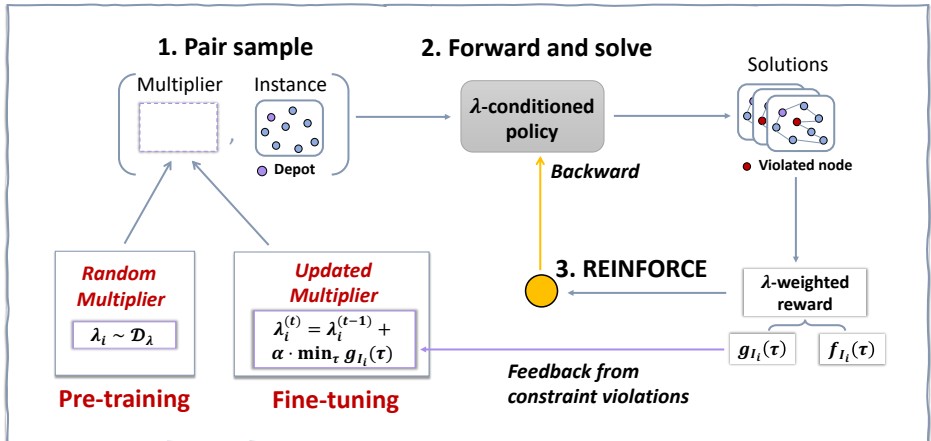

Figure 2: A sketch of the two training stages. In both stages, the $\lambda$-conditioned policy is trained using REINFORCE. The primary distinction lies in the handling of the multiplier $\lambda$. During the pre-training stage, $\lambda$ is randomly sampled to facilitate early convergence and to enhance the policy's adaptability across diverse $\lambda$ values. In contrast, the fine-tuning stage employs an iterative update mechanism for $\lambda$, ensuring that its values are precisely adjusted to account for constraint violations.

the *advantage* that measures relative reward improvement over the shared baseline. Intuitively, the training algorithm reinforces the probability of generating positive advantage trajectories (i.e., solutions) while decreasing the probability of generating negative ones. The pseudo code of the pre-training process is provided in Appendix B. Through this training process with random $\lambda$, the conditioned policy obtains the adaptability to different levels of constraint awareness. Additionally, this pre-training phase ensures sufficient convergence of the policy, effectively reducing the occurrence of infeasible instances to a manageable level. Once these objectives are achieved, the training transitions to the subsequent stage, where instance-specific $\lambda$ values are iteratively optimized.

**Fine-tuning stage.** To achieve an effective alignment between $\lambda$ values and instance hardness, we further fine-tune the pre-trained policy using iteratively updated $\lambda$ values. In this stage, we initialize a uniform and small initial value $\lambda^{(0)}$ for all instances and alternate between optimizing the policy and updating the multipliers. For policy optimization, we continue to employ the REINFORCE algorithm with an average baseline, as used in the pre-training stage. For updating the multipliers, the subgradient is computed based on the minimal constraint violation value across a set of sampled solutions $\{\tau^j\}_{j=1}^P$. Formally, the $\lambda$ values are updated by the following rule:

$$\lambda_i^{(t)} = \lambda_i^{(t-1)} + \alpha \min_{j \in [P]} (g_{I_i}(\tau^j) + c_{I_i}(\tau^j)),$$

where $\alpha$ is the learning rate. After each iteration, we retain the infeasible instances and their corresponding $\lambda$ values in the batch while replacing the feasible instances with new ones. It is important to note that the pre-trained policy is already capable of finding feasible solutions for the majority of instances. Therefore, the proportion of infeasible instances in each batch is typically small. Moreover, to further enhance training efficiency and avoid excessive focus on particularly hard instances, we impose a maximum iteration limit and a cap on the infeasible instance ratio. The pseudo code of the fine-tuning process is provided in Appendix B.

**Network architecture.** The $\lambda$-conditioned policy solves instances with a controllable level of constraint awareness, determined by the condition variable $\lambda$. Similar conditioned policies have been explored in related works, particularly for multi-objective optimization [44, 66] and latent space search [13]. Among them, there are two possible ways to incorporate the target variable into the policy network: (1) embedding it into the initial input features or (2) embedding it into the decoder's context. In this paper, we adopt the $\lambda$-conditioned initial embedding, which empirically demonstrates superior performance in adjusting trade-off behaviors (see Appendix F.4). Specifically, building on the POMO model [41], we incorporate a linear transformation of $\lambda$ into the original initial embeddings. The embedding is computed as:

$$\boldsymbol{h}_i^{(0)} = W^\lambda \lambda + W^h [x_i, y_i, l_i, u_i]^\top,$$

where $W^\lambda \in \mathbb{R}^{d \times 1}$ and $W^h \in \mathbb{R}^{d \times 4}$ are trainable parameters, and $[x_i, y_i, l_i, u_i]$ represents the concatenation of the node's coordinates $(x_i, y_i)$ and its time window bounds $(l_i, u_i)$. This concatenated feature vector serves as the input representation for each node. The output $\boldsymbol{h}_i^{(0)}$ is then used as the initial embedding for the encoder network, which employs the multi-head attention mechanism [63] to perform message passing and update node embeddings. Intuitively, the $\lambda$-conditioned embedding adjusts the relative importance of distance-related features (e.g., node coordinates) and constraint-related features (e.g., time window bounds) based on the value of $\lambda$, thereby enabling a controllable level of constraint awareness. The rest of the architecture closely follows the standard model [41].

# 4 Experiments

In this section, we evaluate the effectiveness of our ICO method through comparison experiments and ablation studies. Additional results are included in Appendix F due to space limitation.

## 4.1 Experimental Settings

**Problem instance generation.**   Following prior works [40], we randomly sample node coordinates $(x_i, y_i)$ from a uniform distribution $U(0, 1)$ within a square. For generating the time windows of TSPTW and draft limits of TSPDL, we utilize the code from Bi et al. [9] and adopt the **hard** settings, which are sufficiently challenging to examine state-of-the-art neural and OR solvers.

**Implementation details.**   Our model is implemented based on the POMO framework [41], incorporating the PI mask [9] to restrict the search space. We only employ the PIP decoder to predict masks during the training process on instances with the number of nodes $n = 100$. The prior distribution of $\lambda$ in the pre-training stage, i.e., $\mathcal{D}(\lambda)$, is set to a triangular distribution $T(0.1, 0.5, 2.0)$. The learning rate for updating $\lambda$ is set to $0.5$ for TSPTW and $0.2$ for TSPDL. The common hyperparameters shared between our method and prior works follow their default settings [41, 9]. In evaluation, our method employs $\times 8$ instance augmentation and 16 iterations to update $\lambda$ during the inference stage. To align the runtime consumption, we use sampling strategies for PIP. More implementation details are provided in Appendix E due to space limitation.

**Baselines.**   We compare our proposed method against state-of-the-art neural methods and OR solvers. For OR solvers, we include LKH3 [30], one of the strongest solver specifically designed for VRPs; and OR-Tools [20], a general-purpose solver capable of handling various constraints. For neural methods, we consider the state-of-the-art PIP framework [9]. For TSPTW(DL)100, we report the results of the models with the PIP decoder. Our experiments **encompass four configurations of PIP**: $\lambda = 0.5$, $\lambda = 1.0$, $\lambda = 2.0$, and a dynamically updated $\lambda$. Specifically, in the dynamic setting, the value of $\lambda$ is periodically adjusted using subgradient ascent every 1000 epochs. The subgradient is estimated based on the average constraint violation observed on the validation dataset.

**Metrics.**   We evaluate performance and efficiency using four metrics: infeasibility rate, average optimality gap, normalized HyperVolume (HV) and runtime. Among these, the HV serves as a comprehensive indicator, capturing both feasibility and solution quality. A detailed explanation of these metrics is provided in Appendix E.3.

## 4.2 Main Results

**Comparison with single-$\lambda$ models.**   The performance comparisons on TSPTW and TSPDL across different problem scales are presented in Table 1. On TSPTW100, the proposed ICO method reduces the infeasibility rate from $4.33\%$ (achieved by POMO+PIP with $\lambda = 1.0$) to an impressive $1.33\%$, representing a substantial reduction of $3.00\%$. Similarly, on TSPTW50, the infeasibility rate is lowered from $1.56\%$ to just $0.50\%$. Even when the $\lambda$ value in single-$\lambda$ models is increased to $2.0$, these models still lag behind the ICO method in terms of feasibility, with the sole exception being TSPDL100. In addition to improving feasibility rates, the ICO method consistently outperforms single-$\lambda$ models in terms of optimality gaps. For instance, the ICO method achieves a smaller gap of $9.22\%$ on TSPDL100, compared to $10.77\%$ achieved by the best POMO+PIP model. Moreover, the ICO method showcases the highest HV scores on all benchmarks, indicating its superior trade-off performance in balancing solution quality and constraint satisfaction.

Table 1: Experimental results on TSPTW and TSPDL. Test instances are generated using the hard settings [9]. LKH3 (less time) and OR-Tools (less time) denote the OR methods with reduced runtime budgets to align with neural solvers. The best and the runner-up results are highlighted in **Blue** and **Violet**, respectively.

| Methods | TSPTW50 | | | | TSPTW100 | | | |
|---|---|---|---|---|---|---|---|---|
| | Inf. Rate ↓ | Avg. Gap ↓ | HV ↑ | Time ↓ | Inf. Rate ↓ | Avg. Gap ↓ | HV ↑ | Time ↓ |
| LKH3 | 0.12% | 0.0% | 1.00 | 7h | 0.07% | 0.0% | 1.00 | 1.4d |
| OR-Tools | 65.72% | 0.0% | 0.34 | 2.4h | 89.07% | 0.0% | 0.11 | 1.6d |
| LKH3 (less time) | 57.34% | **0.01%** | 0.43 | 100s | 95.56% | **0.03%** | 0.04 | 8m |
| OR-Tools (less time) | 65.72% | **0.02%** | 0.34 | 99s | 89.07% | 0.51% | 0.10 | 8m |
| AM + PIP ($\lambda = 1.0$) | 2.99% | 0.34% | 0.90 | 105s | 7.80% | 0.70% | 0.79 | 8m |
| POMO + PIP ($\lambda = 0.5$) | 1.95% | 0.08% | **0.96** | 108s | 4.90% | 0.17% | **0.92** | 9m |
| POMO + PIP ($\lambda = 1.0$) | 1.56% | 0.16% | 0.95 | 108s | **4.33%** | 0.25% | 0.91 | 9m |
| POMO + PIP ($\lambda = 2.0$) | 1.41% | 0.19% | 0.95 | 108s | 4.71% | 0.39% | 0.88 | 9m |
| POMO + PIP (dynamic $\lambda$) | **0.98%** | 0.13% | 0.93 | 108s | 4.94% | 0.45% | 0.87 | 9m |
| ICO (Ours) | **0.50%** | 0.07% | **0.98** | 91s | **1.33%** | **0.14%** | **0.96** | 8m |

| Methods | TSPDL50 | | | | TSPDL100 | | | |
|---|---|---|---|---|---|---|---|---|
| | Inf. Rate ↓ | Avg. Gap ↓ | HV ↑ | Time ↓ | Inf. Rate ↓ | Avg. Gap ↓ | HV ↑ | Time ↓ |
| LKH3 | 0.0% | 0.0% | 1.00 | 6.8h | 0.0% | 0.0% | 1.00 | 1.2d |
| OR-Tools | 100.0% | / | / | 10.6s | 100.0% | / | / | 56.8s |
| LKH3 (less time) | 7.42% | 4.23% | 0.20 | 70s | 7.02% | **6.76%** | 0.20 | 6m |
| OR-Tools (less time) | 100.0% | / | / | 3s | 100.0% | / | / | 29s |
| POMO + PIP ($\lambda = 0.5$) | 3.44% | 2.36% | 0.58 | 71s | 62.94% | 20.95% | / | 5m |
| POMO + PIP ($\lambda = 1.0$) | 1.18% | **2.33%** | 0.78 | 71s | 3.23% | 10.77% | 0.31 | 5m |
| POMO + PIP ($\lambda = 2.0$) | **0.12%** | 2.89% | **0.85** | 71s | 0.11% | 12.24% | **0.38** | 5m |
| POMO + PIP (dynamic $\lambda$) | 0.13% | 2.99% | 0.84 | 71s | **0.01%** | 14.78% | 0.26 | 5m |
| ICO (Ours) | **0.01%** | **2.32%** | **0.88** | 69s | 0.91% | **9.22%** | **0.49** | 5m |

**Comparion with strong OR solvers.** In Table 1, we also compare our neural methods with strong OR solvers, LKH3 and OR-Tools, under aligned runtime conditions. The results show that our ICO method achieves a dramatic improvement in infeasibility rates, reducing them from 95.56% to 1.33% (94.23% reduction) on TSPTW100 and from 7.02% to 0.91% (6.11% reduction) on TSPDL100. Regarding solution quality, our method significantly outperforms OR-Tools on TSPTW100 and even surpasses LKH3 on TSPDL50. While the solution quality of our neural approach on the other three benchmarks still lags behind LKH3, the substantial improvements in feasibility and competitive performance overall underscore the strengths of our neural method.

**Comparison under different inference strategies.** In Table 2, we extend the scope of our comparative experiments to incorporate additional inference strategies, including *Greedy* (vs. $T = 1$), *Sampling* (vs. $T > 1$), and *Efficient Active Search (EAS)* [32]. The results consistently demonstrate that ICO outperforms the best-performing PIP model (denoted as PIP*) in most scenarios. In particular, our ICO integrates well with *EAS*, achieving near-zero infeasibility rates and gaps. The only exception on ICO ($T = 1$) can be attributed to the small initial $\lambda$ value. Notably, we observe that ICO ($T = 2$) even surpasses PIP* (*Sampling* 16) while consuming much less runtime, which highlights the superiority of our proposed ICO. To defense the prolonged runtime of ICO, we further compare ICO ($T = 16$) with LKH3 post search. The results indicate that even adding a strong post-search such as LKH3 to the baseline, our ICO method remains superior in reducing infeasibility.

**Analysis of anytime performance.** During inference, our ICO method iteratively samples solutions and updates $\lambda$, making the anytime performance throughout the iterative process a critical factor. Figure 3 shows the convergence curves of infeasibility rate and average optimality gap on TSPTW50 and TSPTW100. The results indicate that, while ICO starts with a higher infeasibility rate, it converges rapidly and outperforms single-$\lambda$ models in later iterations. In terms of optimality gap, ICO consistently achieves better results throughout the process.

**Extension to more problem variants.** Our proposed method can be seamlessly extended to solve more VRP variants. In Appendix F.1, we conduct comparison experiments on two kinds of CVRPTW. The results show that our proposed ICO method still have advantages on CVRPTW.

Table 2: Comparisons under different inference strategies, including *Greedy*, *Sampling*, *Efficient Active Search (EAS)* [32] and *LKH3 post search*. When *EAS* is integrated, the number of parallel solutions (i.e., POMO size) is increased to 10 for estimating the average baseline. We select the best PIP model from the four configurations according to the HV metric, denoted by PIP*. For LKH3 post search, the solutions generated by PIP*(*Greedy*) are used as the initial solutions of LKH3. *Sampling/EAS* 16 refers to conducting 16 iterations, while $T$ represents the iteration count of our ICO method. The best and the runner-up results are highlighted in **Blue** and **Violet**, respectively.

| Methods | TSPTW50 (10k instances) | | | | TSPTW100 (1k instances) | | | |
|---|---|---|---|---|---|---|---|---|
| | Inf. Rate ↓ | Avg. Gap ↓ | HV ↑ | Time ↓ | Inf. Rate ↓ | Avg. Gap ↓ | HV ↑ | Time ↓ |
| PIP* (*Greedy*) | 3.05% | 0.22% | 0.927 | 9s | 9.00% | 0.23% | 0.868 | 4s |
| PIP* (*Sampling* 2) | 2.53% | 0.10% | 0.955 | 14s | 7.20% | 0.22% | 0.887 | 7s |
| PIP* (*Sampling* 16) | 2.11% | 0.09% | 0.961 | 63s | 5.80% | 0.19% | 0.906 | 43s |
| PIP* (*EAS* 16) | 1.22% | 0.05% | 0.978 | 11m | **0.50%** | 0.04% | **0.987** | 9m |
| **PIP* (*Greedy*) + LKH3** | 1.40% | **0.01%** | **0.984** | 100s | 6.17% | **-0.07%** | 0.951 | 50s |
| ICO ($T=1$) | 2.14% | 0.10% | 0.959 | 9s | 14.10% | 0.15% | 0.833 | 4s |
| ICO ($T=2$) | 1.67% | 0.09% | 0.966 | 14s | 3.60% | 0.17% | 0.931 | 7s |
| ICO ($T=16$) | **0.50%** | 0.07% | **0.981** | 90s | 1.10% | 0.14% | 0.961 | 48s |
| ICO (*EAS* 16) | **0.17%** | **0.03%** | **0.993** | 11m | **0.20%** | **0.02%** | **0.994** | 9m |

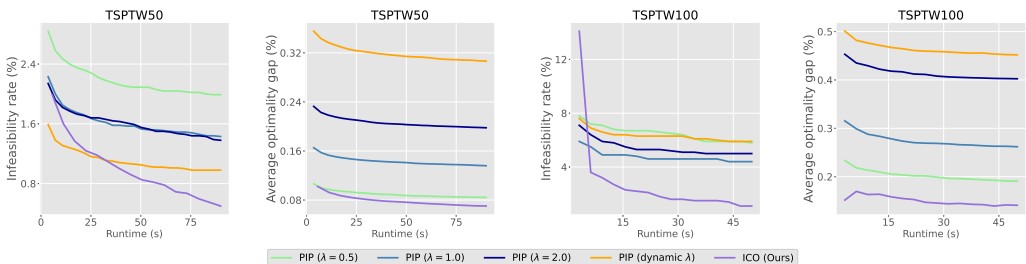

Figure 3: Anytime performance comparion between our ICO method and the single-$\lambda$ methods.

## 4.3 Ablation Study

In this subsection, we present a series of experiments to investigate the impact of each component. Detailed results and analyses are provided in Appendix F due to space limitation.

- **Analysis of update rules for $\lambda$ in inference stage.** See Appendix F.2.
- **Analysis of training strategies.** See Appendix F.3.
- **Analysis of the $\lambda$-conditioned network architecture.** See Appendix F.4.
- **Analysis of the pre-defined $\lambda$ distribution.** See Appendix F.5.
- **Sensitivity analysis of $\lambda$-related hyperparameters.** See Appendix F.6.

## 5 Conclusion

In this paper, we propose a novel approach ICO to address the limitations of existing Lagrangian-based neural methods in solving complex constained VRPs. Unlike prior methods that rely on a single, uniform multiplier across all problem instances, ICO leverages instance-specific multipliers to improve adaptability and better optimize the trade-off between solution quality and constraint satisfaction for every problem instance. Experimental results on two challenging constrained VRP benchmarks, TSPTW and TSPDL, demonstrate that ICO significantly reduces infeasibility rates compared to both state-of-the-art neural methods and strong OR solvers like LKH3. These empirical findings suggest that our ICO framework can be a promising alternative for strong OR solvers when tackling constrained combinatorial problems. **One limitation of this study** lies in the fact that the proposed ICO framework necessitates a minimum of two iterations to update the $\lambda$ values, resulting in an extended inference runtime. Future research could explore methods for directly predicting optimal $\lambda$ values, improving the training strategies of the conditioned policy, and enabling generalization across diverse sets of constraints.

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

# A  Illustration of our proposed method

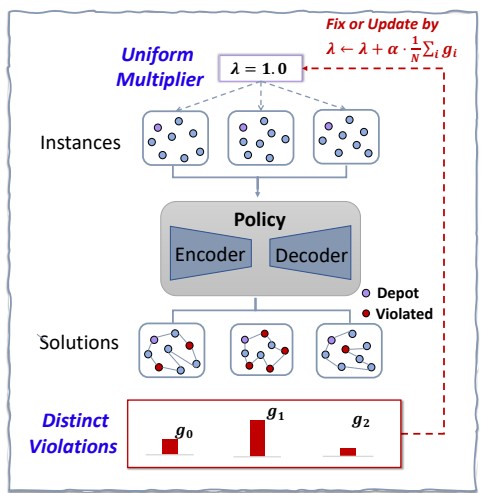 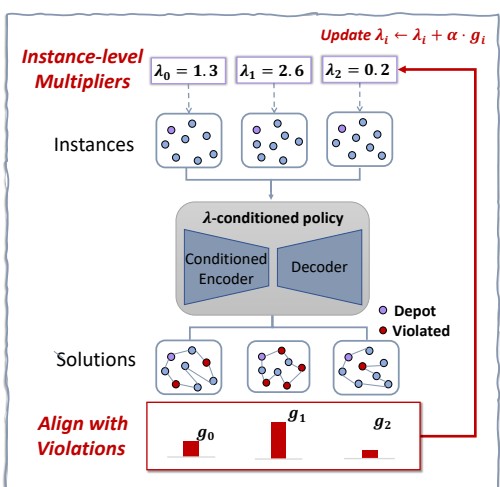

**(a) Previous policy-level multiplier methods**    **(b) Our proposed instance-level adaptive method**

Figure 4: An illustration is presented to compare previous *policy-level* multiplier methods with our proposed *instance-level* adaptive approach. The *policy-level* methods are limited in their ability to address distinct constraint violations across diverse instances, as they apply uniform multipliers irrespective of instance-specific variations. In contrast, our *instance-level* method inherently aligns multiplier values with the specific constraint violations of each instance, thereby achieving more precise and adaptive handling of constraints.

# B  Pseudo Code of the Training Process

---

**Algorithm 1** Pre-training of the $\lambda$-conditioned policy

---

**Input:** Distribution $D_\lambda$, number of batches $T$, batch size $B$, number of parallel sampling $P$
Initialize policy network parameters $\theta$
**for** $t = 0$ **to** $T - 1$ **do**
    Generate a batch of instances $\{I_i\}_{i=1}^B$
    Sample multipliers $\lambda_i \sim D_\lambda, \quad \forall i \in \{1, ..., B\}$
    Sample multiple solutions $\{\tau_i^j\}_{j=1}^P \sim \pi_\theta(\cdot | \lambda_i, I_i), \quad \forall i \in \{1, ..., B\}$
    Compute baseline $b_i \leftarrow \frac{1}{P} \sum_{j=1}^P -(f_{I_i}(\tau_i^j) + \lambda_i(g_{I_i}(\tau_i^j) + c_{I_i}(\tau_i^j))), \quad \forall i \in \{1, ..., B\}$
    Compute policy gradient $\nabla_\theta J(\theta) \leftarrow \frac{1}{BP} \sum_{i=1}^B \sum_{j=1}^P (-(f_{I_i}(\tau_i^j) + \lambda_i(g_{I_i}(\tau_i^j) + c_{I_i}(\tau_i^j))) - b_i) \nabla_\theta \log \pi_\theta(\tau_i^j | \lambda_i, I_i)$
    Update parameters $\theta \leftarrow \theta + \alpha \nabla_\theta J(\theta)$
**end for**
**Output:** $\theta$

---

# C  Related Works

**Prevalent paradigms of neural VRP.**    Many researchers have focused on end-to-end neural methods that learn to generate solutions through deep neural networks [6, 11]. These neural solvers can be categorized into three paradigms [50]: (1) **Learn-to-Construct (L2C) methods** sequentially extends solutions from scratch in an autoregressive manner, typically trained via reinforcement learning [51] or imitation learning [19]. These L2C methods have proven to be applicable to a variety of combinatorial problems [70] and industrial applications [42]. (2) **Learn-to-Predict (L2P) methods** operate under a variable-independent assumption, directly predicting the entire solution

---

**Algorithm 2** Fine-tuning of the $\lambda$-conditioned policy

---

**Input:** Number of batches $T$, batch size $B$, number of parallel sampling $P$, multiplier learning rate $\alpha_\lambda$, policy learning rate $\alpha$, maximum number of itertations $K$, maximum infeasible ratio $\delta$

Initialize policy network parameters $\theta$

Generate a batch of instances $\{I_i\}_{i=1}^B$

Initialize multipliers $\lambda_i \leftarrow \lambda^{(0)}, \quad \forall i \in \{1, ..., B\}$

Initialize iteration counts $k_i \leftarrow 0, \quad \forall i \in \{1, ..., B\}$

**for** $t = 0$ **to** $T - 1$ **do**

    Sample multiple solutions $\{\tau_i^j\}_{j=1}^P \sim \pi_\theta(\cdot|\lambda_i, I_i), \quad \forall i \in \{1, ..., B\}$

    Compute baseline $b_i \leftarrow \frac{1}{P} \sum_{j=1}^P -(f_{I_i}(\tau_i^j) + \lambda_i(g_{I_i}(\tau_i^j) + c_{I_i}(\tau_i^j))), \quad \forall i \in \{1, ..., B\}$

    Compute policy gradient $\nabla_\theta J(\theta) \leftarrow \frac{1}{BP} \sum_{i=1}^B \sum_{j=1}^P (-(f_{I_i}(\tau_i^j) + \lambda_i(g_{I_i}(\tau_i^j) + c_{I_i}(\tau_i^j))) - b_i)\nabla_\theta \log \pi_\theta(\tau_i^j|\lambda_i, I_i)$

    Update parameters $\theta \leftarrow \theta + \alpha \nabla_\theta J(\theta)$

    Adjust the maximum number of iterations $K$ according to the current infeasibility ratio, ensuring the ratio of retained infeasible instances does not exceed the maximum ratio $\delta$

    **for** each instance $I_j$ without feasible solutions **do**

        Update $\lambda_j \leftarrow \lambda_j + \alpha_\lambda \min_{m \in [P]}(g_{I_j}(\tau_j^m) + c_{I_j}(\tau_j^m))$

        Increment $k_j \leftarrow k_j + 1$

    **end for**

    **for** each instance $I_j$ with zero $k_j$ or $k_j > K$ **do**

        Generate a new instance to replace $I_j$

        Initialize $\lambda_j \leftarrow \lambda^{(0)}$ and $k_j \leftarrow 0$

    **end for**

**end for**

**Output:** $\theta$

---

without conditional dependence [35]. While computationally efficient, L2P methods often suffer from limited expressiveness. To address this issue, recent research has introduced diffusion models to enhance the L2P paradigm by leveraging their ability to generate multimodal distributions of optimal solutions [59, 43]. (3) **Learn-to-Search (L2S) methods** adopt the iterative framework of traditional search heuristics. During the search process, L2S methods usually leverage a RL policy to control or select search operators [49, 46], thereby guiding the search directions towards near-optimal solutions.

**Recent advances in neural VRP.** Recent advancements in neural methods for solving VRPs focus on improving scalability and robustness through innovative architectures and learning strategies. For example, the large-scale performance is improved by employing divide-and-conquer strategies [23, 69], leveraging heavy decoder architectures [47], incorporating distance-related bias [73], and exploiting local transferability [24, 22]; the robustness against distribution shifts is improved by distributional robust optimization [33], multi-distribution knowledge distillation [8], meta learning [74] and ensemble learning [34]. Furthermore, it is observed that the performance of neural solvers can be enhanced by utilizing a population of complementary models [27, 76, 25]. Moreover, Liu et al. [45] proposed to develop a foundation model for a class of VRP variants, leveraging the shared problem structure to achiece better performance. Building on this, Zhou et al. [75] further improved model capability by introducing the mixture-of-experts structure. Besides these efforts, this paper focuses on complex constrained VRPs, which are common in real-world applications [12, 26] but have not received much attention in the research community. Only a few works [60, 15, 9] try to address it through feature enhancement or Lagrange multiplier method. In this context, we introduce a novel instance-level adpative framework for Lagrangian-based neural methods, reducing the infeasiblity rate significantly.

**Learning for constrained optimization in other domains.** Most neural solvers for constrained optimization problems rely on expert-designed rules to prevent constraint violations during the decoding process [3, 59, 36]. For instance, Ahn et al. [3] proposed a clean-up phase that rolls back invalid actions, thereby enforcing feasibility through a hard constraint mechanism. However, these expert-driven rules often fail in complex scenarios, such as the constrained VRPs studied in this work, as well as in many continuous optimization problems. To address constraint violations in such cases,

optimization techniques based on penalty functions and Lagrange multipliers have been integrated into unsupervised learning [2] and self-supervised learning pipelines [53]. However, a common limitation arises when a single multiplier or penalty factor is applied across diverse problem instances with varying degrees of constraint violations. This challenge, though important, has been largely overlooked in prior studies [2, 65, 54]. Only a related study on continuous optimization [53] noticed this issue and introduced a primal-dual network to predict instance-specific dual variables during training. While this approach improves upon the single multiplier method, it remains constrained by the computational overhead associated with alternating primal-dual ascent and the limited accuracy of dual variable predictions.

## D  Instance Generation

In our experiments, we consider two categories of problem, TSPTW and TSPDL. Following prior works [40], we randomly sample coordinates $(x_i, y_i)$ for each node $i$ (including the depot) from a uniform distribution $U(0,1)$ within a square. For generating the time windows and draft limits, we utilize the code of Bi et al. [9] and adopt the **hard** settings, which are sufficiently challenging to examine state-of-the-art neural and OR solvers. The generation process of time windows and draft limits is detailed as follows.

**Time windows.**  After generating the node coordinates, the pairwise travel times are calculated based on the Euclidean distance between any two nodes. For the generation of time windows, we adopt the configuration of a widely recognized benchmark [17] in our experiments. Specifically, the process begins with the construction of a random tour $\tau$ (i.e., a random permutation of the nodes). Subsequently, the time window $[l_i, u_i]$ for each node $i$ is iteratively generated, where the lower bound $l_i$ and upper bound $u_i$ are uniformly sampled from a range determined by the cumulative travel distance $\phi_i$ of the partial solution up to node $i$ and the maximum window size $2\eta$. More formally, $l_i \sim U[\phi_i - \eta, \phi_i]$ and $u_i \sim U[\phi_i, \phi_i + \eta]$. This procedure guarantees the existence of at least one feasible solution for each instance, and the tight coupling between the time windows and the randomized tours introduces significant complexity to the problem, thereby increasing the computational difficulty of satisfying constraints. In this paper, the maximum window size $\eta$ is set to 50, and we employ a scale factor $\rho = 100$ to normalize the node coordinates and time windows according to [9].

**Draft limits.**  In the context of TSPDL, each node is associated with a demand value and a maximum draft limit, which is designed to avoid overloaded ships entering these ports (i.e., nodes). From an initial feasible setting, the draft limit of each node is set to the summarized demands of other nodes, thereby ensuring that any node demand can not exceed its own draft limit. Subsequently, a fraction parameter, denoted as $p\%$, is introduced to adjust the draft limits of non-depot nodes. Specifically, $p\%$ of the non-depot nodes are randomly selected, and each of them is assigned a draft limit drawn as a random integer from the range $[\delta_i, \sum_{i=1}^{n} \delta_i]$, where $\delta_i$ is the demand of the $i$-th node. Finally, a feasibility validation is conducted (e.g., utilizing bin-counting constraints) to ensure that the assigned draft limits do not lead to instances without feasible solutions. In our experiment, the node demands are set to 1 and the fraction parameter $p\%$ is set to $90\%$.

## E  Implementation Details

### E.1  Training Details

The training procedure of our ICO method contains two stages: a pre-training stage and a fine-tuning stage. The pre-training stage involves a total of $10,000$ epochs, while the fine-tuning stage comprises $1,000$ epochs. Each training epoch processes $10,000$ synthetic problem instances. For both stages, we select the model checkpoint that achieves the best inference performance on a validation dataset as the final model. It is worth noting that the training process of our ICO method includes $1,000$ more epochs compared to the training process of POMO+PIP. To ensure a fair comparison, we extend the training of the provided POMO+PIP checkpoints by an additional $1,000$ epochs.

The fine-tuning stage involves the iterative updating of $\lambda$ values. In this process, the initial values $\lambda^{(0)}$ is uniformly set to 0.1 for all problem instances. If the policy fails to find feasible solutions

on a specific instance, the $\lambda$ value corresponding to this instance is updated based on the constraint violation, where the learning rate of $\lambda$ is set to $0.5$ for TSPTW and $0.2$ for TSPDL, since the scales of constraint violations on TSPTW and TSPDL are different. These hyperparameters in updating $\lambda$ are aligned with the corresponding hyperparameters in the inference stage, narrowing the gap of training and inference. To improve computational efficiency and mitigate the risk of overfocusing on challenging instances, the number of iterations is limited to a maximum of $4$, and the ratio of infeasible instances within a batch must not exceed $25\%$. During the fine-tuning on TSPDL50, we observe that the fine-tuned policy tends to overemphasize the constraints, resulting in a near zero infeasibility rate but a significant deterioration in objective values. To mitigate this issue, we adjust the learning rate of fine-tuning process on TSPDL50 to $1 \times 10^{-6}$, while learning rates of other training process remain the default setting (i.e., $1 \times 10^{-4}$).

### E.2 Inference Details

The instance-specific $\lambda$ values are iteratively updated based on constraint violations during the inference stage. In this process, the $\lambda$ values are initialized as $0.1$ for all instances except instances of TSPDL100, since it is observed that the conditioned policy fails to obtain feasible solutions for most instances of TSPDL100 when using $\lambda = 0.1$. Consequently, the intial $\lambda$ value for TSPDL100 is increased to $0.5$. During the updating process of $\lambda$, the learning rate is configured as $0.5$ for TSPTW and $0.2$ for TSPDL. These different learning rates are to accommodate the different scales of constraint violations on these two problem types. In the comparison experiments, the number of iterations for updating $\lambda$ is set to $16$.

### E.3 Experimental Settings

**Metrics.** Four metrics are applied: Infeasibility rate, average optimality gap, normalized Hyper-Volume (HV) and runtime. The instance-level infeasibility rate measures the proportion of instances where the solver fails to find any feasible solution. These metrics are calculated on a test dataset containing 10,000 instances. To compute the optimality gap, we use the solutions obtained by LKH3 through full-time search as reference solutions. Unlike some prior works that compute the optimality gap directly from the average objective [40], we calculate the optimality gap on an instance-by-instance basis and then average these values. It is important to note that the calculation of objective values and optimality gaps only includes instances with feasible solutions. Therefore, the average objective value may not serve as a fully reliable metric for performance comparison, as the sets of instances with feasible solutions can vary across different methods. To measure the comprehensive performance of both solution quality and feasibility, we further compute the normalized HV based on the infeasibility rate and average optimality gap. The reference point for computing HV is set to $(100\%, 5\%)$ for TSPTW and $(10\%, 20\%)$ for TSPDL, where the first number represent the infeasibility rate and the other denotes the average gap. To evaluate the computational efficiency, we compare the total runtime of solving 10,000 instances with batch parallelism on a single GPU (NVIDIA RTX 4090 Ti). For OR solvers like LKH3 and OR-Tools, we record the runtime of parallel computation on 16 CPU cores.

**Evalution configurations of baselines.** To align the runtime consumption, POMO+PIP employs $\times 28$ sampling for intances with $n = 50$ and $\times 20$ sampling for instances with $n = 100$, where AM+PIP adopts $\times 200$ sampling for both $n = 50$ and $n = 100$ instances. These different sampling configurations are to align with the additional runtime caused by the computation of $\lambda$-conditioned embeddings in our ICO method. The evaluation batch sizes for both POMO-PIP and our ICO method are set to 2,500 for instances with $n = 50$ and 1000 for instances with $n = 100$.

## F Additional Results

### F.1 Extension to more problem variants.

The idea of instance-level adaptive dual variables is not specially designed for TSPTW and TSPDL; rather, it can be extended to other domains that simultaneously require constraint handling and cross-instance (or cross-environment) generalization of the RL policy, with domain-specific adaptations. To demonstrate generality, we extend our method to more VRP variants. After summarizing the

hard-constrained VRPs addressed in prior works [15, 21, 9, 60, 14], we find that CVRPTW is the only problem not addressed in our experiments. While the decision space of CVRPTW appears more complex, it is, in fact, easier to satisfy its constraints compared to TSPTW and TSPDL. This is because its time window constraints can be easily satisfied by a shortcut: Add more vehicles.

To construct a challenging benchmark, we propose to **set a maximum limit on the number of vehicles**, which also aligns more closely with real-world applications. We conduct new experiments on CVRPTW50 with limited vehicles using JAMPR's time window generation code [21]. Since PIP has not been extended to this problem, we used POMO as the backbone to implement ICO. Experimental results in Table 3 show that our ICO significantly outperforms the POMO baseline, especially in infeasibility rate.

Table 3: Experimental results on new problem variants: CVRPTW50 and CVRPTW50 with limited vehicles. To compute HV, we use reference point (1%, 15) for CVRPTW50 and (10%, 15) for CVRPTW50 with limited vehicles. The best results are highlighted in **bold**.

| | **CVRPTW50** | | | | **CVRPTW50 with limited vehicles** | | | |
|---|---|---|---|---|---|---|---|---|
| Method | Inf. rate | Obj. | HV | Time | Inf. rate | Obj. | HV | Time |
| POMO ($\lambda = 0.5$) | 0.69% | **13.99** | 0.021 | 39s | 4.35% | **14.05** | 0.036 | 38s |
| POMO ($\lambda = 1.0$) | 0.25% | 14.22 | 0.039 | 40s | 3.26% | 14.28 | 0.033 | 38s |
| POMO ($\lambda = 2.0$) | 0.31% | 14.49 | 0.023 | 39s | 2.51% | 14.51 | 0.025 | 38s |
| ICO | **0.10%** | 14.00 | **0.060** | 40s | **1.16%** | 14.09 | **0.054** | 40s |

## F.2 Analysis of Different Update Rules for $\lambda$

**Proportional-Integral-Derivative (PID) control for updating $\lambda$.** From the perspective of control theory, the subgradient ascent process of $\lambda$ behaves as *integral* control, while Stooke et al. [58] proposed to further incorporate *proportional* and *derivative* control into the update rule, avoiding oscillations encountered by the integral-only controller. The proportional control is to hasten the constraint satisfaction in response to the immediate constraint violation. The derivative control prevents the oscillations by monitoring the variation tendency of constraint violations. By adding the terms of proportional, integral and derivative control, the update rule of PID control is expressed as:

$$\Delta_t = g_I(\tau_t),$$
$$I_t = I_{t-1} + g_I(\tau_t),$$
$$\delta_t = \max\{g_I(\tau_t) - g_I(\tau_{t-1}), 0\},$$
$$\lambda_t = K_P \cdot \Delta_t + K_I \cdot I_t + K_D \cdot \delta_t,$$

where $\Delta_t$ represents the proportional term of time step $t$, $I_t$ denotes the $t$-th step integral term that accumulates the constraint violations of previous steps, $\delta_t$ computes the derivative term of the constraint violation, and $K_P, K_I, K_D$ are tuning parameters that measure the weights of three terms. Intuitively, this PID method provides a richer set of controllers than subgradient ascent, but it also introduces more hyperparameters that require manual tuning. In our experiments, $K_P$ is set to 0.1 and $K_D$ is set to 1.0 on both problem types, and $K_I$ is set to 0.5 on TSPTW and 0.01 on TSPDL.

In Table 4, we compare the performance of different update rules of $\lambda$ in inference stage: fixed $\lambda$ values ($\lambda \in \{0.5, 1.0, 2.0\}$), randomly sampled $\lambda$ values, the subgradient ascent method and the PID control method [58]. For the random sampling strategy, $\lambda$ values are drawn randomly from the uniform distribution $U(0.1, 2.0)$ at each iteration.

The results in the last three rows indicate that both the subgradient ascent method and the PID control method generally outperform the random sampling strategy, with particularly improvements in reducing the infeasibility rate. As evidenced in the first three rows, employing fixed $\lambda$ values leads to significantly inferior performance compared to the adaptive variation of $\lambda$, underscoring the critical importance of dynamically adjusting $\lambda$ for each instance. It is worth noting that the random sampling approach also demonstrates competitive performance, indicating that simply varying the $\lambda$ values randomly for each instance has a high probability of identifying effective $\lambda$ values. By comparing

the results of the last two rows, it is observed that the PID control method does not achieve superior performance as expected, which can be attributed to two factors: (1) the hyperparameters of PID are challenging to tune; (2) the subgradient ascent method is already involved in the fine-tuning process, while the PID control is not integrated into the training, limiting its effectiveness.

Table 4: Additional results of different update rules of $\lambda$ on TSPTW and TSPDL. The best results are highlighted in **bold**.

| Methods | TSPTW ($n = 50$) | | TSPTW ($n = 100$) | | TSPDL ($n = 50$) | | TSPDL ($n = 100$) | |
|---|---|---|---|---|---|---|---|---|
| | Inf. rate | Avg. Gap | Inf. rate | Avg. Gap | Inf. rate | Avg. Gap | Inf. rate | Avg. Gap |
| ICO ($\lambda = 0.5$) | 1.43% | 0.19% | 4.34% | 0.26% | 2.63% | 2.50% | 42.14% | 13.16% |
| ICO ($\lambda = 1.0$) | 1.52% | 0.23% | 4.03% | 0.36% | 0.23% | 2.77% | 2.01% | 10.79% |
| ICO ($\lambda = 2.0$) | 1.55% | 0.24% | 4.27% | 0.38% | 0.07% | 3.15% | 0.38% | 11.62% |
| ICO (random) | 0.55% | 0.07% | 2.40% | 0.14% | 0.12% | **2.28%** | 0.40% | 10.73% |
| ICO (subgradient) | **0.51%** | 0.07% | **1.33%** | 0.14% | **0.01%** | 2.32% | 0.91% | **9.22%** |
| ICO (PID control) | 0.55% | 0.07% | 1.39% | 0.14% | 0.05% | 2.36% | **0.26%** | 9.25% |

## F.3 Analysis of training strategies

Figure 5 illustrates the performance of POMO+PIP (with $\lambda = 1$), the pre-trained policy, and the fine-tuned policy. The comparison between the pre-trained and fine-tuned policies reveals that the fine-tuning process leads to a substantial reduction in both infeasibility rate and average gap, except the average gap on TSPDL50. Notably, even the pre-trained policy alone surpasses the single-$\lambda$ POMO+PIP, further highlighting the advantages of the proposed approach.

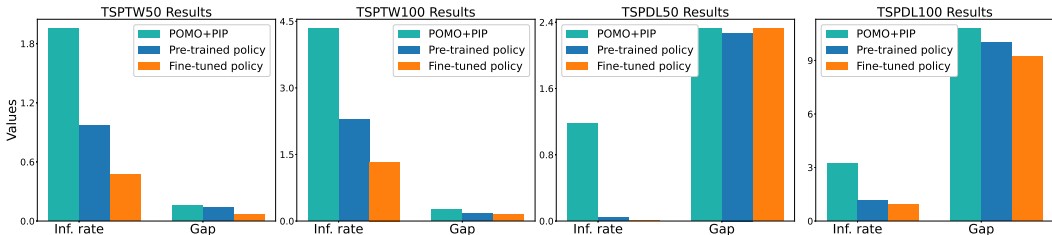

Figure 5: Comparison of the pre-trained policy and fine-tuned policy.

## F.4 Analysis of Network Architectures

The $\lambda$-conditioned policy network takes $\lambda$ as the condition varibale and adjust the constraint awareness according to the varying value of $\lambda$. Among existing network architectures in other domains [66, 44], there are two alternative approaches to implement the conditioned policy: (1) condition $\lambda$ in the initial embeddings; (2) condition $\lambda$ in the decoder's context. The second approach, referred to as the $\lambda$-conditioned context method, is detailed as follows.

$\lambda$**-conditioned context.** Building upon the POMO model [41], the conditioned context method integrates a linear embedding of $\lambda$ into the decoder's *context* embedding, formulated as $\boldsymbol{q} = W^\lambda \lambda + W^q[\boldsymbol{h}^c, t^c]$. Here, $W^\lambda \in \mathbb{R}^{d \times 1}$ and $W^q \in \mathbb{R}^{d \times d}$ are trainable parameters, and $[\boldsymbol{h}^c, t^c]$ denotes the concatenation of the current node embedding $\boldsymbol{h}^c$ and the current time $t^c$, together forming the *context* used for selecting candidate nodes. The resulting output, $\boldsymbol{q}$, functions as the query input for the subsequent multi-head attention layer in the decoder. This conditioned context approach incorporates the information of $\lambda$ into the core component of the decoder, enabling an efficient adjustment of the policy's behavior.

In Table 5, we compare the performance of the network with $\lambda$-conditioned context and network with $\lambda$-conditioned embeddings on TSPTW100 and TSPDL100. Here we report the results of the pre-trained policies. The experimental results demonstrate that the $\lambda$-conditioned embedding method achieves significantly superior performance in both infeasibility rate and average optimality gap. This performance advantage can be attributed to the fact that the $\lambda$-conditioned embedding utilizes the

Table 5: Additional results of different network architectures on TSPTW and TSPDL. The best results are highlighted in **bold**.

| Methods | TSPTW ($n = 100$) | | TSPDL ($n = 100$) | |
|---|---|---|---|---|
| | Inf. rate | Avg. Gap | Inf. rate | Avg. Gap |
| Network with $\lambda$-conditioned context | 2.83% | 0.30% | 2.31% | 13.34% |
| Network with $\lambda$-conditioned embeddings | **2.28%** | **0.17%** | **1.14%** | **10.01%** |

full capacity of the entire network to process $\lambda$-related information, while the conditioned context approach restricts the $\lambda$-related information to the decoder, thereby limiting its effectiveness.

### F.5 Analysis of the distribution $D(\lambda)$ in training stage

In the pre-training stage of the conditioned policy, random values of $\lambda$ are sampled from a pre-defined distribution $D(\lambda)$ for training. Empirically, the distribution $D(\lambda)$ has a non-negligible influence on the performance of the pre-trained policy. A natural and straightforward option for $D(\lambda)$ is the uniform distribution within an appropriate range. However, as shown in Figure 6, the trained policy just silghtly violates constraints on the majority of instances, where only a small subset of instances in the long tail experience significant constraint violations. Therefore, we adopt a triangular distribution $T(0.1, 0.5, 2.0)$, which biases the sampling towards smaller $\lambda$ values, thereby prioritizing the optimization of instances with low constraint violations. Figure 7 compares the performance of the policy trained with a uniform distribution $U(0.1, 2.0)$ and the policy trained with a triangular distribution $T(0.1, 0.5, 2.0)$ on the TSPTW50 dataset. The results demonstrate that the triangular distribution leads to superior overall performance as expected.

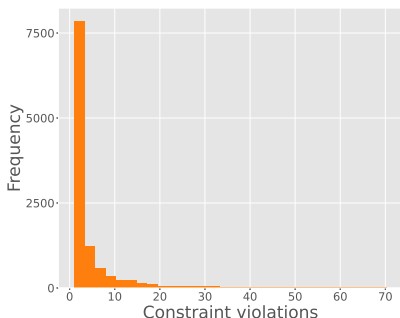

Figure 6: Histogram of constraint violation statistics on the validation dataset.

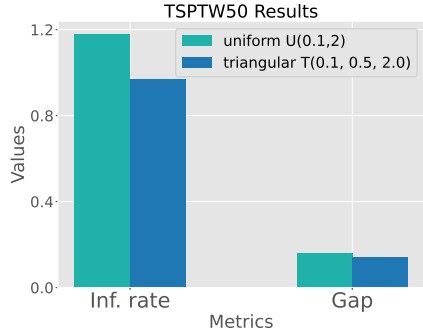

Figure 7: Performance of using two different $D(\lambda)$ configurations during the pre-training stage.

### F.6 Sensitivity of $\lambda$-related hyperparameters in inference stage

Since the optimization landscape for $\lambda$ is typically non-convex due to the hardness of combinatorial optimization, the initial value and learning rate of $\lambda$ are both important for the optimization performance. Here we conduct a sensitivity analysis of $\lambda$ from these two perspectives, including the initial value of $\lambda$ (denoted as $\lambda_0$) and the learning rate for updating $\lambda$ (denoted as $\alpha$). During the inference stage, we evaluated performance across $\lambda_0 \in \{0.1, 0.15, 0.20, 0.5, 1.0\}$ and $\alpha \in \{0.1, 0.2, 0.5, 0.7, 1.0\}$ on TSPTW50 and TSPDL50. Each hyperparameter was varied while keeping the other fixed at its default value. Results in Table 6 and 7 show that:

- In 16 out of 18 settings, our ICO method surpasses the best-performing PIP model in hypervolume (HV), showing its robustness.

- Although the performance variance (shown in the last row) is relatively small, it is not negligible. This underscores the importance of carefully tuning $\lambda$-related hyperparameters to achieve optimal performance.

Interestingly, some settings (e.g., $\alpha = 0.7$ for TSPTW50) slightly outperform the default, suggesting that advanced hyperparameter optimization techniques could further enhance performance.

Table 6: Sensitivity analysis in inference stage on TSPTW50. $\lambda_0$ denotes the initial value of $\lambda$ and $\alpha$ represents the learning rate of $\lambda$.

|  | Inf. rate | Gap | HV | Better HV than PIP |
|---|---|---|---|---|
| PIP with the best HV | 1.95% | 0.08% | 0.965 | - |
| $\lambda_0 = 0.1$ (default) | 0.50% | 0.07% | 0.981 | Yes |
| $\lambda_0 = 0.15$ | 0.47% | 0.08% | 0.979 | Yes |
| $\lambda_0 = 0.2$ | 0.48% | 0.08% | 0.979 | Yes |
| $\lambda_0 = 0.5$ | 0.84% | 0.19% | 0.954 | No |
| $\lambda_0 = 1.0$ | 0.97% | 0.23% | 0.945 | No |
| $\alpha = 0.1$ | 0.59% | 0.07% | 0.980 | Yes |
| $\alpha = 0.2$ | 0.49% | 0.07% | 0.981 | Yes |
| $\alpha = 0.5$ (default) | 0.50% | 0.07% | 0.981 | Yes |
| $\alpha = 0.7$ | 0.48% | 0.07% | 0.981 | Yes |
| $\alpha = 1.0$ | 0.55% | 0.07% | 0.981 | Yes |
| Avg $\pm$ Std | 0.59% $\pm$ 0.17% | 0.10% $\pm$ 0.06% | 0.974 $\pm$ 0.0133 | - |

Table 7: Sensitivity analysis in inference stage on TSPDL50. $\lambda_0$ denotes the initial value of $\lambda$ and $\alpha$ represents the learning rate of $\lambda$.

|  | Inf. rate | Gap | HV | Better HV than PIP |
|---|---|---|---|---|
| PIP with the best HV | 0.12% | 2.89% | 0.845 | - |
| $\lambda_0 = 0.1$ (default) | 0.01% | 2.32% | 0.883 | Yes |
| $\lambda_0 = 0.15$ | 0.01% | 2.33% | 0.883 | Yes |
| $\lambda_0 = 0.2$ | 0.01% | 2.33% | 0.883 | Yes |
| $\lambda_0 = 0.5$ | 0.01% | 2.51% | 0.874 | Yes |
| $\lambda_0 = 1.0$ | 0.01% | 2.79% | 0.859 | Yes |
| $\alpha = 0.1$ | 0.06% | 2.23% | 0.883 | Yes |
| $\alpha = 0.2$ (default) | 0.01% | 2.32% | 0.883 | Yes |
| $\alpha = 0.5$ | 0.00% | 2.47% | 0.877 | Yes |
| $\alpha = 0.7$ | 0.00% | 2.58% | 0.871 | Yes |
| $\alpha = 1.0$ | 0.00% | 2.80% | 0.860 | Yes |
| Avg $\pm$ Std | 0.01% $\pm$ 0.02% | 2.47% $\pm$ 0.20% | 0.876 $\pm$ 0.009 | - |

 # G    Licenses

Table 8: List of licenses for the codes and datasets we used in this work.

| Resource | Type | Link | License |
|---|---|---|---|
| OR-Tools [20] | Code | https://github.com/google/or-tools | Apache License 2.0 |
| LKH3 [30] | Code | http://webhotel4.ruc.dk/ keld/research/LKH-3/ | Available for academic research use |
| AM [40] | Code | https://github.com/wouterkool/attention-learn-to-route | MIT License |
| POMO [41] | Code | https://github.com/yd-kwon/POMO | MIT License |
| EAS [32] | Code | https://github.com/ahottung/EAS | Available online |
| JAMPR [21] | Code | https://github.com/jokofa/JAMPR | MIT License |
| PIP [9] | Code | https://github.com/jieyibi/PIP-constraint | MIT License |

