# OpenReview forum: "Adaptive Constrained Optimization for Neural Vehicle Routing"
_NeurIPS.cc/2025/Conference — Submitted to NeurIPS 2025_

### Official Review · Reviewer_5viD · 2025-06-12

**Clarity:** 3
**Significance:** 3
**Originality:** 3
**Rating:** 5
**Confidence:** 3

**Summary:**

This paper proposes an two-stage adaptive constraint optimization method, which reexpresses the Lagrange dual problem of combinatorial optimization problems by assigning its own multipliers to each specific instance. The algorithm has achieved better results on two special cases with more constraints in the TSP problem.

**Questions:**

See Weaknesses. Mainly Weakness 1. If you inform me of the specific formula of HV, or if you give the performance of the source data based on multiple distributions or real datasets like TSPLIB, I may improve my score.
Small issue:
1.What is the full name of PIP? Please give its full name when first appear in the text, no matter whether it is the content of the important literature you have cited or not.

**Ethical Concerns:**

["NO or VERY MINOR ethics concerns only"]

**Final Justification:**

As the author have addressed my concerns, I decide to raise my score.

**Limitations:**

Yes.

**Paper Formatting Concerns:**

Although the author has conducted sufficient ablation experiments, if possible, the comparison contents and results of the ablation experiments should be briefly introduced in the main text to enhance the readability of the main text.

**Quality:**

3

**Strengths And Weaknesses:**

Strength:
1.This paper raises a profound issue. The coefficient design of the Lagrange multiplier method depends on each problem, and this paper proposes a method to solve it, which is of great significance.
2.The author completed sufficient ablation experiments, proving that multiple designs in the proposed model are important.
3.The proposed network architecture is easy to read.
Weakness:
1.It seems that the article does not provide the specific calculation method of HV, which has raised some doubts in me about the validity of HV. Except for the TSPDL50 problem where the feasible Rate and Average Gap cannot achieve simultaneous SOTA, if there is no convincing HV formula, the persuasiveness of this comparison is insufficient.
2.The scale of the questions for comparison is limited. Increasing the scale of the problem, adding the types of sampling, and adopting real datasets such as TSBLIB may better solve this problem.

---

> ### Author Rebuttal · Authors · 2025-07-31
>
> Thanks for dedicating your time to reviewing our paper! Your suggestions are very constructive for us to further improve the paper. Please find our detailed response below.
>
> ## Response to “… the formulation of HV …”
>
> Thank you for your insightful and constructive comment. In our experiments, we employ the standard normalized hypervolume (HV) metric [1,2]. Its formulation on two-objective minimization problems is given by:
>
> $\text{HV}(\mathbf{f}) = \prod_{j=1}^{2} \left( \frac{z_j^{\mathrm{ref}} - f_j}{z_j^{\mathrm{ref}} - z_j^{\mathrm{ideal}}} \right)$
>
> where $(f_1, f_2)$ denote the two objective values, i.e., the infeasibility rate and the optimality gap in our experiments. The terms $(z_1^{\text{ref}}, z_2^{\text{ref}})$ are the reference values, while $(z_1^{\text{ideal}}, z_2^{\text{ideal}})$ represent the ideal values, which are set to zero in our experiments.
>
> This normalized HV metric effectively captures the overall optimization performance by simultaneously reflecting both feasibility and optimality. Since the two objectives have comparable numerical scales, their contributions to the HV value are approximately balanced. As a result, even when improvements are made in only one aspect (e.g., on TSPDL100), the HV metric remains capable of distinguishing the better-performing methods.
>
> We sincerely appreciate your helpful comment and hope this explanation addresses your concern.
>
> [1] Multicriteria Optimization. Springer, 2005.
> [2] Performance assessment of multiobjective optimizers: An analysis and review. IEEE TEvC, 2003
>
> ## Response to “... the scale of the questions for comparison is limited ...”
>
> Thank you for your constructive and valuable suggestions. Although TSPTW (DL) with $N = 100$ and $N=50$ already poses significant challenges to existing approaches, we fully agree that the problem scale in our current experiments is relatively limited.
>
> In response, we extend our evaluation to larger-scale and real-world datasets by integrating the proposed ICO method with the recently developed **RELD** model [3], which is specifically designed for cross-instance generalization. Both RELD and RELD + ICO are trained on instances with $N \sim \mathcal{U}[40, 70]$, and evaluated on larger-scale datasets, including **TSPTW200-medium** and **real-world TSPLIB** instances. This setup enables a more comprehensive assessment of the generalization ability of our method under substantial shifts in problem scale and distribution.
>
> ### Generalization to TSPTW200 (medium difficulty)
>
> |  | Inf. Rate | Avg. Gap | HV  | Time |
> | --- | --- | --- | --- | --- |
> | POMO | 90.90% | 36.02% | 0.025 | 54s |
> | PIP | 0.00% | 26.04% | 0.479 | 316s |
> | RELD | 4.20% | 21.49% | 0.546 | 55s |
> | RELD + ICO | 0.00% | **21.30%** | **0.574** | 54s |
>
> ### Generalization to TSPLIB ($N\le 200$, medium difficulty)
>
> |  | Inf. rate | Avg. cost | Time |
> | --- | --- | --- | --- |
> | POMO | 0.00% | 21.61 | 85s |
> | PIP | 0.00% | 21.79 | 215s |
> | RELD | 0.00% | 22.10 | 96s |
> | RELD + ICO | 0.00% | **21.47** | 96s |
>
> Note that the backbones POMO, PIP, and RELD are all trained with Lagrangian relaxation. These experimental results summarized above demonstrate that:
> 1. RELD + ICO consistently outperforms other methods in terms of both optimality and feasibility on larger-scale and real-world datasets, highlighting the robustness and scalability of the proposed approach.
> 2. Compared to PIP, which requires computing a mask with **cubic time complexity** in the number of nodes, our method can eliminate the need for such expensive computation on large-scale problems. In our experiments on TSPTW200, RELD + ICO achieves **a significant reduction in inference time (316s to 54s)** while maintaining higher solution quality and ensuring 0% infeasibility.
>
> We will revise our paper to include these new experiments and analysis. Thank you again for your valuable suggestion, which has helped us further strengthen our experimental design.
>
> [3] Rethinking Light Decoder-based Solvers for Vehicle Routing Problems, ICLR 2025.
>
> ## Response to minor issues
>
> Thank you for your kind reminder. “PIP” refers to *Proactive Infeasibility Prevention*. We will revise the paper to make this abbreviation clear. As for the ablation experiments, we will consider shrinking our methodology and experiment setup sections to leave more space for those interesting experiments. Thank you again for your valuable suggestions!
>
> ---
>
> **We hope that our response has addressed your concerns, but if we missed anything, please let us know.**

---

> > ### Author Response · Authors · 2025-08-08
> >
> > Dear Reviewer 5viD,
> >
> > We hope this message finds you well.
> >
> > We are writing to kindly follow up on the rebuttal we submitted. In response to your valuable suggestions, we have conducted additional experiments on larger-scale problems and real-world benchmarks (i.e., TSPLIB), and provided a more detailed explanation of the HV formulation to address your concerns regarding the evaluation metrics.
> >
> > As the discussion phase is concluding soon (by tomorrow), we just wanted to ensure you had a chance to review our response and share any further thoughts, if you wish.
> >
> > We sincerely appreciate the time and effort you’ve dedicated to reviewing our work, and we completely understand how busy schedules can be. Thank you once again for your thoughtful feedback and consideration.
> >
> >
> > Best regards,
> >
> > authors

---

### Official Review · Reviewer_D8c6 · 2025-06-19

**Clarity:** 3
**Significance:** 2
**Originality:** 3
**Rating:** 4
**Confidence:** 4

**Summary:**

This paper builds upon the PIP model introduced in "Learning to Handle Complex Constraints for Vehicle Routing Problems", with a focus on addressing routing challenges such as TSPTW and TSPDL, where masks alone cannot easily prevent infeasible solutions. The key advancement lies in iteratively adjusting Lagrangian multipliers at the instance level within the PIP framework. This approach enhances the optimization of the dual objectives of solution feasibility and objective function value. By employing subgradient methods, the paper introduces an innovative solution to improve upon the fixed Lagrangian multipliers in the original PIP model.

**Questions:**

1. **Comparison with PIP-D**: The original paper "Learning to Handle Complex Constraints for Vehicle Routing Problems" also proposed a PIP-D model, which achieved state-of-the-art results in most cases. Why does this paper not include comparisons with PIP-D? This omission may leave a gap in understanding the full scope of the model's performance.
2. **Training Efficiency**: During training, the model introduces an additional parameter lambda to enhance the base model's generalization ability regarding lambda. To strengthen the credibility of the proposed model, it would be valuable to present:
   - The training time required for a single lambda to achieve comparable effectiveness to the PIP model.
   - The training time needed for multiple different lambdas of the base model to match the performance of the PIP model.
   - Comparative experiments between the base model and the enhanced model to provide a comprehensive assessment of training efficiency and effectiveness.
3. **Generalization During Inference**: During inference or fine-tuning, as lambda iterates, if a lambda value not encountered during training appears, does the model possess sufficient capacity to adapt and learn effectively?

**Ethical Concerns:**

["NO or VERY MINOR ethics concerns only"]

**Final Justification:**

The authors have made significant improvements to address the reviewer's prior concerns:

1. **Enhanced Generalization and Efficiency**: The authors supplemented experiments on the model's generalization capabilities and large-scale efficiency, which helps to strengthen the credibility of the results.

2. **Clarified PIP-D and Inference Generalization Mechanisms**: The authors provided a more detailed elaboration on PIP-D and the generalization mechanism during inference, effectively resolving the reviewer's doubts on these points.

Although the paper's contribution to complex-constrained problems remains relatively limited, its perspective offers valuable insights into solving combinatorial optimization problems with complex constraints. Consequently, the reviewer has decided to increase the score by 1 point.

**Limitations:**

More executing time may be required during the inference.

**Quality:**

2

**Strengths And Weaknesses:**

## Strengths
1. The paper is well-organized, following a logical progression from motivation to analysis, methodology, and experimental validation. This structure significantly enhances readability and comprehension.
2. The proposed method demonstrates notable improvements in learning-based approaches for problems like TSPTW and TSPDL, where constraints cannot be directly managed through masks. This advancement broadens the applicability of learning-based methods in complex routing scenarios.

## Weaknesses
1. The paper's innovation primarily revolves around the dynamic iterative improvement and adjustment of fixed Lagrangian multipliers in the PIP model. This singular focus may limit the perceived novelty of the contribution.
2. While the method shows promise, it does not consistently outperform the PIP model. Specifically, on the TSPTW50 dataset, the objective function value offers no advantage, and on the TSPDL100 dataset, instance feasibility shows no improvement. Furthermore, the lack of comparison with the PIP model on CVRP datasets, despite acknowledged implementation challenges, weakens the overall persuasiveness of the results. The paper also fails to demonstrate significant superiority on other benchmark datasets.
3. The paper does not adequately explore the model's generalization capabilities, leaving questions about its applicability to a wider range of problems or variations.

---

> ### Author Rebuttal · Authors · 2025-07-31
>
> Thanks for dedicating your time to reviewing our paper! Your suggestions are very constructive for us to further improve the paper. Please find our detailed response below.
>
> ## Response to “... adjustment of fixed Lagrangian multipliers … This singular focus may limit the perceived novelty of the contribution”
>
> Thank you for your valuable and constructive feedback. Targeting important constrained combinatorial problems with broad applications, the contributions of this paper have two folds:
>
> 1. **Conceptual Contribution**: To the best of our knowledge, we are the first to reveal the limitations of using a policy-level Lagrangian multiplier (i.e., a single $\lambda$) in neural combinatorial optimization. This insight can help to rectify a common but suboptimal practice and, we believe, contribute meaningfully to further development of this research direction.
> 2. **Technical Contribution**: Building on this conceptual insight, we propose a sound instance-level multiplier formulation. We design an effective multiplier-conditioned policy and successfully incorporate the iterative Lagrangian algorithm into both training and inference. Extensive experiments and ablation studies demonstrate the effectiveness of each component of our method.
>
> While our method may appear relatively simple—without introducing novel architectures or complex algorithms—we emphasize that the primary contribution of this work lies in its conceptual advancement. We believe this perspective can inspire future research in constrained VRP and encourage a more principled use of Lagrangian training methods, beyond fixed penalty weights. In fact, the simplicity of our approach enables a model-agnostic framework, which can be readily applied to other RL-based backbones such as pure POMO and RELD [1] (see the next response). Thank you again for your valuable feedback. We will revise our paper to include more discussions and clarifications.
>
> [1] Rethinking Light Decoder-based Solvers for Vehicle Routing Problems, ICLR 2025.
>
> ## Response to “... it does not consistently outperform the PIP model... ”
>
> Thank you very much for your valuable and constructive feedback. We would like to begin by revisiting the experimental results. Across the four datasets (TSPTW50/100 and TSPDL50/100) and three evaluation metrics (Infeasibility Rate, Gap, and HV), our proposed ICO method outperforms all PIP baselines in **11 out of 12** cases. The only exception is the feasibility rate on TSPDL100. While we acknowledge that some improvements are modest, we believe the overall performance gain is sufficient to justify the contribution and effectiveness of our method.
>
> In addition, the superiority of our proposed ICO extends beyond incremental performance gain in the following two key aspects:
>
> 1. **Model-Agnostic Framework:** Our method is not tailored specifically to the PIP model; rather, it can be integrated into a wide range of RL-based backbones. To demonstrate this, we conducted additional experiments combining ICO with both pure POMO and RELD [1]—the latter being a recent method designed for cross-instance generalization with a more complex decoder. Following the generalization setting of RELD, we trained both RELD and RELD + ICO on instances with node size $N \sim \mathcal{U}[40,70]$, and evaluated them on the larger TSPTW200-medium dataset, which allows us to assess the generalization capability under a shift in problem scale. The results summarized below show that ICO consistently improves the performance over POMO and RELD, even on unseen larger instances, underscoring its effectiveness and general applicability across different architectures.
>
> 2. **Efficiency on solving large-scale problems:** Compared to PIP, which requires computing a mask with **cubic time complexity** in the number of nodes, our method can eliminate the need for such expensive computation in some cases. For example, on large instances with $N=200$, RELD + ICO achieves **a significant reduction in inference time (316s to 54s)** while maintaining high solution quality and ensuring 0% infeasibility, as shown in the last table.
>
> ### Combined with POMO on TSPTW50 (Medium Difficulty)
>
> | Method | Inf. Rate | Avg. Gap | HV | Time |
> | --- | --- | --- | --- | --- |
> | POMO (sample 2) | 1.91% | 4.87% | 0.42 | 7s |
> | POMO + ICO (T = 2) | **0.86%** | **4.68%** | **0.49** | 8s |
> | POMO (sample 28) | 0.68% | 3.90% | 0.57 | 47s |
> | POMO + ICO (T = 16) | **0.19%** | **3.56%** | **0.63** | 47s |
>
> ### Combined with POMO on TSPDL50 (Medium Difficulty)
>
> | Method | Inf. Rate | Avg. Gap | HV | Time |
> | --- | --- | --- | --- | --- |
> | POMO (sample 2) | 12.18% | 3.57% | 0.25 | 7s |
> | POMO + ICO (T = 2) | **1.68%** | 4.61% | **0.49** | 8s |
> | POMO (sample 28) | 11.25% | 2.91% | 0.31 | 47s |
> | POMO + ICO (T = 16) | **0.19%** | 3.47% | **0.65** | 47s |
>
> ### Combined with RELD and generalization to TSPTW200 (Medium Difficulty)
>
> |  | Inf. Rate | Avg. Gap | HV  | Time |
> | --- | --- | --- | --- | --- |
> | POMO | 90.90% | 36.02% | 0.025 | 54s |
> | PIP | 0.00% | 26.04% | 0.479 | 316s |
> | RELD | 4.20% | 21.49% | 0.546 | 55s |
> | RELD + ICO | 0.00% | **21.30%** | **0.574** | 54s |
>
> We sincerely appreciate your comments again and hope this response adequately addresses your concerns.
>
> ## Response to “… explore the model’s generalization capabilities …”
>
> To the best of our knowledge, prior constrained optimization methods for VRP have primarily focused on TSPTW, CVRPTW, and TSPDL. As summarized in the table below, our work is comparatively more comprehensive in its coverage of these variants:
>
> | Related work | TSPTW | TSPDL | CVRPTW |
> | --- | --- | --- | --- |
> | MUSLA (Chen et al., 2024) | Yes | No | No |
> | JAMPR (Falkner & Schmidt-Thieme, 2020) | No | No | Yes |
> | Chen et al., 2022 | No | No | Yes |
> | Tang et al., 2022 | Yes | No | Yes |
> | PIP (Bi et al., 2024) | Yes | Yes | No |
> | Ours | Yes | Yes | Yes |
>
> If we apply the ICO method to problems beyond VRP or even combinatorial optimization, extensive domain-specific adaptations are necessarily required. However, the core idea of instance-level multiplier formulation is inherently general and may be applicable to other settings. In any RL scenario that requires both constraint satisfaction and generalization across instances, the limitations of policy-level multipliers also apply, making ICO a potentially valuable solution. We sincerely appreciate your comments again and hope this response adequately addresses your concerns.
>
> ## Response to “Comparison with PIP-D”
>
> Thank you for your thoughtful feedback. **As noted in lines 260–261, “For TSPTW (DL) 100, we report the results of the models with the PIP decoder”**, we adopt the PIP decoder (PIP-D) for TSPTW (DL) 100. Based on the results of PIP's paper, which show that PIP-D improves performance on $N=100$ datasets, but the original PIP has better results on $N=50$ datasets, we report results using PIP on $N=50$ and PIP-D on $N=100$ to reflect the best performance of PIP models. For simplicity, we refer to both variants under the unified “PIP” in our paper. We are sorry for this confusion, and will revise to make it clearer.
>
> ## Response to “Training efficiency”
>
> Thanks for your valuable question. First, we want to clarify that the training times of the PIP method and the proposed ICO method over 110k epochs **are approximately the same**: for example, around 60 hours for TSPTW50 and about 10 days for TSPTW100. At first glance, fine-tuning $\lambda$ appears computationally intensive. However, in our implementation, we mitigate this by using a fixed number of batches, and in each batch, only the feasible instances are replaced with new ones, as described in lines 216-217. That is, if there is an increasing number of infeasible instances requiring iterative updates in the current batch, the number of newly generated instances in future batches will decrease correspondingly. As a result, the total number of network forward and backward computations remains equivalent to that of the original PIP method.
>
> Additionally, ICO's training metrics don’t directly reflect iterative inference, making the sample efficiency hard to evaluate during training. Instead, **we compare methods under aligned training budgets**, as we did in Tables 1 and 2.
>
> **In response to Point 1**, we conducted new experiments using fixed $\lambda$ to train our model. The results show that using a fixed $\lambda$ in our model yields no difference from the baseline under the same training time.
>
> ### Fixed $\lambda$ results on TSPTW50
>
> |  | Inf. rate | Avg. gap | HV |
> | --- | --- | --- | --- |
> | PIP ($\lambda=1$, 60h training) | 1.95% | 0.08% | 0.96 |
> | ICO (fixed $\lambda=1$, 60h training) | 1.94% | 0.08% | 0.96 |
>
> **Regarding Points 2 and 3**, please refer to the results of PIP (dynamic $\lambda$) in Tables 1 and 2 of our paper. **PIP (dynamic $\lambda$) applies subgradient ascent to adjust the single $\lambda$**, serving as an advanced form of the “multiple $\lambda$” model you suggested. The results in Tables 1 and 2 show that our ICO method consistently outperforms this dynamic version, highlighting the effectiveness of instance-level $\lambda$ adaptation.
>
> ## Response to “Generalization during inference”
>
> Good point! Empirically, ICO generalizes reasonably well across varying $\lambda$ values. During training, the sampled or updated $\lambda$ values roughly range from 0.1 to 2.0. At test time, ICO handles $\lambda$ up to 5.0 for most instances, and up to 10.0 for some with large violations, showing robustness to unseen values. However, when $\lambda$ becomes excessively large or small (e.g., 0.0001), model performance may degrade or even collapse. While improving this aspect may further enhance performance, the current model’s generalization ability has already been sufficient to effectively optimize most instances.
>
> ---
>
> **We hope that our response has addressed your concerns, but if we missed anything, please let us know.**

---

> > ### Comment · Reviewer_D8c6 · 2025-08-06
> >
> > The authors have made significant improvements to address the reviewer's prior concerns:
> >
> > 1. **Enhanced Generalization and Efficiency**: The authors supplemented experiments on the model's generalization capabilities and large-scale efficiency, which helps to strengthen the credibility of the results.
> >
> > 2. **Clarified PIP-D and Inference Generalization Mechanisms**: The authors provided a more detailed elaboration on PIP-D and the generalization mechanism during inference, effectively resolving the reviewer's doubts on these points.
> >
> > Although the paper's contribution to complex-constrained problems remains relatively limited, its perspective offers valuable insights into solving combinatorial optimization problems with complex constraints. Consequently, the reviewer has decided to increase the score by 1 point.

---

> > > ### Author Response · Authors · 2025-08-06
> > >
> > > Thanks for your feedback! We are glad to hear that your concerns have been addressed. We will make sure to include the added results and discussions in the final version. Thank you.

---

### Official Review · Reviewer_JxWh · 2025-06-30

**Clarity:** 3
**Significance:** 2
**Originality:** 2
**Rating:** 4
**Confidence:** 3

**Summary:**

The paper introduces ICO, an optimization framework designed to address the limitations of traditional NCO methods, where the Lagrange multipliers are fixed. Experimental results demonstrate that ICO yields promising results, particularly for the TSPTW and TSPDL.

**Questions:**

1. Can the authors provide experiments of ICO under the other two difficulty levels, as PIP does?

2. The paper uses the "instance infeasibility rate" but does not consider "solution infeasibility rate." Could the authors explain why the latter was excluded?

3. Can ICO be applied to other algorithms based on Lagrange multipliers? A discussion on this would clarify its broader applicability.

4. How does the training time of ICO compare to PIP?

**Ethical Concerns:**

["NO or VERY MINOR ethics concerns only"]

**Final Justification:**

The authors have provided additional experiments during the rebuttal, which partially address my concerns. However, similar experimental settings have already been presented in the most relevant works. I remain skeptical about the motivation for not following conventional experimental setups initially and only presenting such results during the limited rebuttal period. I take a neutral position, neither supporting acceptance nor rejection.

**Limitations:**

N/A.

**Paper Formatting Concerns:**

N/A.

**Quality:**

3

**Strengths And Weaknesses:**

**Strengths:**
1. The paper enhances the constraint handling capability of neural VRP solvers, which is an important contribution to the field.
2. The motivation for the proposed method is well-articulated, and it is coherently integrated with the methodology, providing a clear rationale for the approach.
3. The paper is well-structured and easy to follow, with a logical flow between sections.

**Weakness:**
1. The method is primarily based on the specific model PIP, and does not seem to demonstrate generality for different models (e.g., BQ and GOAL). Furthermore, the performance gain (compared to PIP) achieved is not significant.

2. As the experiments (include three difficulty levels) conducted in PIP paper, the evaluation of this work is only conducted under the "hard" difficulty level, which limits the generalizability of the findings.

3. The experimental section lacks clarity in certain areas. For example, the default setting for λ in POMO+PIP and how it was configured for the TSPTW / DL of size 50 are not specified.

4. The proposed approach is heuristic in nature, and there is no theoretical guarantee or proof of its effectiveness, which raises concerns about its long-term reliability and robustness.

---

> ### Author Rebuttal · Authors · 2025-07-31
>
> Thanks for dedicating your time to reviewing our paper! Your suggestions are very constructive for us to further improve the paper. Please find our detailed response below.
>
> ## Response to “... does not seem to demonstrate generality for different models... ”
>
> Thank you for your insightful and constructive feedback. Although our primary experiments are conducted based on the PIP backbone, we would like to clarify that the proposed ICO framework is inherently general and not restricted to any specific model architecture. In fact, it can be naturally extended to many backbone models within the domain of neural combinatorial optimization, as long as these models can be trained via reinforcement learning (RL) and Lagrangian relaxation. When adapting our method, users can simply incorporate the multiplier-conditioned embedding into the input of any neural policy network and employ the sampled or iteratively updated instance-specific multipliers in the RL training pipeline.
>
> In response to your valuable suggestion, we have conducted additional experiments using a recently proposed backbone: **RELD** [1], which is designed to enhance cross-instance generalization through an improved decoder and achieve impressive performance. Your recommendation of BQ and GOAL is reasonable, but their supervised training is not compatible with the Lagrangian-based method. Following the generalization setting of RELD, we trained both RELD and RELD + ICO on instances with node size $N \sim \mathcal{U}[40,70]$, and evaluated them on the larger **TSPTW200-medium** dataset and **TSPLIB** dataset.
>
> ### Combined with RELD and generalization TSPTW200 (medium difficulty)
>
> |  | Inf. Rate | Avg. Gap | HV  | Time |
> | --- | --- | --- | --- | --- |
> | POMO | 90.90% | 36.02% | 0.025 | 54s |
> | PIP | 0.00% | 26.04% | 0.479 | 316s |
> | RELD | 4.20% | 21.49% | 0.546 | 55s |
> | RELD + ICO | 0.00% | **21.30%** | **0.574** | 54s |
>
> ### Combined with RELD and generalization to TSPLIB ($N\le 200$, medium difficulty)
>
> |  | Inf. Rate | Avg. cost | Time |
> | --- | --- | --- | --- |
> | POMO | 0.00% | 21.61 | 85s |
> | PIP | 0.00% | 21.79 | 215s |
> | RELD | 0.00% | 22.10 | 96s |
> | RELD + ICO | 0.00% | **21.47** | 96s |
>
> These results show that RELD + ICO consistently improves feasibility and optimality over RELD, confirming the generality and effectiveness of our method. Importantly, in contrast to PIP—which involves computing a mask with **cubic time complexity** relative to the number of nodes—RELD + ICO circumvents this computational overhead in large-scale instances, achieving a substantial reduction in inference time (from 316s to 54s) while preserving high solution quality and ensuring 0% infeasibility.
>
> We hope these additional results and clarifications address your concerns.
>
> [1] Rethinking Light Decoder-based Solvers for Vehicle Routing Problems. ICLR 2025.
>
> ## Response to “... only conducted under the hard difficulty level …” and “… experiments of ICO under the other two difficulty levels...”
>
> Thank you very much for your constructive and insightful comments.
>
> As evidenced in the experimental results reported by PIP [2] (see Table 1 and Table 2), the infeasibility rates at the instance level on the *easy* datasets are already 0%, and those on the *medium* datasets are below 1%, which indicate that existing neural solvers can already handle such cases with high effectiveness. Therefore, we only selected the hard datasets in our experiments.
>
> Nevertheless, we acknowledge that assessing performance across a broader spectrum of difficulty levels is also valuable. In response to your suggestion, we have conducted additional experiments on medium-level datasets to further examine the effectiveness of our method, as presented below. Experiments on easy datasets are not reported since all the compared neural solvers achieve 0% infeasibility. To make a distinguishable comparison, we used the original POMO as the backbone, instead of PIP.
>
> ### TSPTW50 (Medium Difficulty)
>
> | Method | Inf. Rate | Avg. Gap | HV | Time |
> | --- | --- | --- | --- | --- |
> | POMO (sample 2) | 1.91% | 4.87% | 0.42 | 7s |
> | POMO + ICO (T = 2) | **0.86%** | **4.68%** | **0.49** | 8s |
> | POMO (sample 28) | 0.68% | 3.90% | 0.57 | 47s |
> | POMO + ICO (T = 16) | **0.19%** | **3.56%** | **0.63** | 47s |
>
> ### TSPDL50 (Medium Difficulty)
>
> | Method | Inf. Rate | Avg. Gap | HV | Time |
> | --- | --- | --- | --- | --- |
> | POMO (sample 2) | 12.18% | 3.57% | 0.25 | 7s |
> | POMO + ICO (T = 2) | **1.68%** | 4.61% | **0.49** | 8s |
> | POMO (sample 28) | 11.25% | 2.91% | 0.31 | 47s |
> | POMO + ICO (T = 16) | **0.19%** | 3.47% | **0.65** | 47s |
>
> We can observe that ICO consistently achieves significantly lower infeasibility rates and higher HV scores compared to POMO across medium-level datasets. Notably, in the TSPDL50 medium dataset, ICO reduces the infeasibility rate from over 11% (POMO) to just 0.19% when using T = 16, which further highlights the effectiveness of our method. We will revise our paper to include these new results and analysis.
>
> ## Response to “... the experimental section lacks clarity in certain areas...”
>
> Thank you for your valuable feedback. We respectfully clarify that the experimental settings in our paper are, to our understanding, adequately described. Many details are provided in Section 4.1, Appendix E, and in our released code. Reviewer dvDR has explicitly acknowledged the sufficiency of our provided experimental details. As for your specific concern on $\lambda$ settings, we describe the $\lambda$ configurations of PIP in Section 4.1 (lines 261–264), including $\lambda = 0.5$, $1.0$, $2.0$, and a dynamic variant, which is also clearly presented in Table 1. If any other part of the experimental setup remains unclear, please let us know. We’re happy to provide further clarification.
>
> ## Response to “... there is no theoretical guarantee or proof of its effectiveness...”
>
> Thank you very much for your insightful and constructive feedback. To the best of our knowledge, rigorous theoretical analysis for neural combinatorial optimization methods is still an open and challenging problem in the area. Consequently, a direct theoretical comparison is almost infeasible.
>
> However, we are able to provide a simple justification for our proposed instance-level Lagrangian formulation. Let us define the policy-level objective as $F(\theta,\lambda) =\sum_{i=1}^N\left[ J(\pi_\theta, I_i) +\lambda\cdot J_C(\pi_\theta, I_i)\right]$ and the instance-level objective as $G(\theta,\{\lambda_i\}) =\sum_{i=1}^N\left[ J(\pi_\theta, I_i) +\lambda_i\cdot J_C(\pi_\theta, I_i)\right]$, then we have the following inequality:
>
> $\min_{\{\lambda_i \geq 0\}} \max_\theta G(\theta, \{\lambda_i\}) \leq \min_{\lambda_1=\cdots=\lambda_N =\lambda} \max_\theta G(\theta, \{\lambda_i\})=\min_{\lambda \geq 0} \max_\theta F(\theta, \lambda)$
>
> This result clearly demonstrates that the proposed instance-level formulation is guaranteed to be no worse than the conventional policy-level formulation. Moreover, it also implies that the greater the heterogeneity among the optimal multipliers ${\lambda_i}$, the more expressive power and potential benefit the instance-level formulation is likely to offer.
>
> We hope this clarification helps to address your concern, and we are grateful again for your valuable feedback. In the future, we will put more efforts in theoretical analysis of NCO methods, to fill in the gap in the area.
>
> ## Response to “... uses the instance infeasibility rate but does not consider the solution infeasibility rate.”
>
> Thank you for your insightful comment. The instance-level infeasibility rate measures the fraction of problem instances with no feasible solution found, directly reflecting the method’s ability to satisfy constraints. In contrast, the solution-level rate counts the proportion of infeasible solutions overall. While useful in some cases, it may be misleading here, as it depends heavily on the distribution: easy instances may yield many feasible solutions, while hard ones may yield none. This can lead to a deceptively low solution-level rate even when the method fails on harder instances.
> Given this, we believe the instance-level rate offers a more meaningful and reliable metric for evaluating constraint satisfaction in our study. We will revise to add some discussion to make it clearer. Thank you.
>
> ## Response to “Can ICO be applied to other algorithms based on Lagrange multipliers?”
>
> Thank you very much for your insightful question. We believe your question can be understood from two perspectives:
>
> 1. Whether ICO can be integrated with other algorithms that optimize Lagrange multipliers;
> 2. Whether ICO can be applied to Lagrangian-based methods in broader domains.
>
> Both interpretations raise important and interesting directions for exploration. Regarding the first perspective, ICO is indeed compatible with various Lagrangian multiplier algorithms. For instance, it can be combined with PID control-based approaches, as discussed in Appendix F.2. For the second, we believe ICO has potential in other RL domains requiring constraint satisfaction and cross-instance generalization. In such settings, the limitations associated with policy-level multipliers also exist, making ICO a potentially valuable solution.
>
> ## Response to “How does the training time of ICO compare to PIP?”
>
> Thanks for your valuable question. The training times of the PIP method and the proposed ICO method over 110k epochs **are approximately the same**: for example, around 60 hours for TSPTW50 and about 10 days for TSPTW100. At first glance, fine-tuning $\lambda$ appears computationally intensive. However, in our implementation, we control the number of forward and backward passes by reducing the new instances at the fine-tuning stage, ensuring that the overall training cost remains similar.
>
> ---
>
> **We hope that our response has addressed your concerns, but if we missed anything, please let us know.**

---

> > ### Comment · Reviewer_JxWh · 2025-08-06
> > **Reply to the rebuttal**
> >
> > Thank you for your feedback. After reviewing the authors' responses, I still have the following concerns:
> > 1.  I disagree with the authors' point on the easy and medium difficulty levels. While existing solvers have achieved near-zero infeasibility rates at easy and medium levels, the gap, especially at 100 nodes, still offers room for improvement. TSPTW/DL optimization requires balancing both infeasibility and gap, which remains a key challenge in VRPs with complex constraints. In addition, the authors did not provide results for the easy difficulty setting.
> >
> > 2. The newly provided medium setting results are missing the PIP baseline, which is essential for evaluating any improvements from ICO.
> >
> > 3. The results of PIP and PIP-D were not reported or compared separately, which is necessary for a thorough evaluation.
> >
> > Therefore, I will keep my score.

---

> > > ### Author Response · Authors · 2025-08-07
> > >
> > > Thank you for taking the time to engage in further discussion. We fully agree that the trade-off between infeasibility and optimality is a core challenge in solving complex VRPs. Our proposed ICO method is designed to address this issue through a more principled instance-level Lagrangian optimization framework, which enables the model to adaptively balance feasibility and objective quality on a per-instance basis, rather than solely optimizing the expected policy performance across the dataset.
> > >
> > > However, we would like to respectfully clarify that our experiments on the hard and medium datasets have already demonstrated the superior trade-off capability of our proposed method. In these settings, the constraints are sufficiently complex to influence the optimization process, thereby necessitating a balance between feasibility and objective value. In contrast, for the easy datasets, most solutions generated by the PIP model already lie within the feasible region (in our experiments, the infeasibility rate on TSPTW50-easy quickly drops to 0% within the first few epochs), making the training process primarily focus on minimizing the objective value, with minimal trade-off involved. Therefore, we do not think that the easy setting provides a better benchmark for evaluating the trade-off ability of our method.
> > >
> > > Regarding the reporting of PIP and PIP-D, we would like to clarify our rationale. According to the original PIP paper, PIP-D consistently outperforms PIP only on the N=100 datasets, while for N=50, PIP often performs comparably or even better. To provide the strongest and most representative baselines for each setting, we chose to report PIP for N=50 and PIP-D for N=100. This choice also helps reduce redundancy and improves clarity in our presentation.
> > >
> > > We hope our clarification can address your remaining concerns. Nevertheless, we acknowledge that incorporating the additional experiments you suggested would provide a more comprehensive view, and we are working to include these results. Specifically, we plan to expand our experimental scope to cover:
> > >
> > > 1. Evaluation on the easy setting,
> > >
> > > 2. Additional results using the PIP backbone, and
> > >
> > > 3. Separate comparisons of PIP and PIP-D.
> > >
> > > However, due to the high computational cost of training models with the PIP mask (over 60 hours for N=50 and more than 10 days for N=100), the experiments are currently underway, and we will report the results for N=50 as soon as they are available (before the discussion deadline). Thank you again for your thoughtful comments.

---

> > > > ### Comment · Reviewer_JxWh · 2025-08-09
> > > > **Reply to the rebuttal**
> > > >
> > > > Thank you for the rebuttal.
> > > >
> > > > Firstly, in the newly provided medium-difficulty experimental results, PIP is not included as a comparison baseline, despite its code being publicly available. Therefore, from Table 1, I can only conclude that ICO slightly outperforms PIP in the hard scenario, but no such conclusion can be drawn for other scenarios.
> > > >
> > > > Secondly, realistic settings should cover problems of varying difficulty. Without including experiments on easy, medium, and hard cases, the results remain unconvincing.
> > > >
> > > > Thirdly, although you note that the early infeasibility rate shows low in the easy setting, this does not guarantee that it will remain stable during future optimization, nor that the performance gap will steadily decrease. These two metrics may fluctuate, as we previously discussed; balancing them remains an important research focus.
> > > >
> > > > Considering that the experiments are insufficient, I still keep my score.

---

> > > > > ### Author Response · Authors · 2025-08-09
> > > > > **(1/2) Response**
> > > > >
> > > > > Thank you for taking the time to engage in further discussion.
> > > > >
> > > > > ## Response to "...ICO slightly outperforms PIP in the hard scenario..."
> > > > >
> > > > > We would like to respectfully clarify that the performance improvement of our method over PIP should not be considered "slight". In Table 1, we include four variants of the PIP model, each trained with a different $\lambda$ setting—specifically, $\lambda = 0.5$, $1.0$, $2.0$, and a dynamic $\lambda$—to ensure a comprehensive comparison. We guess that you compared the best result among these four PIP models to our single ICO model, and then concluded that the improvement is slight. However, such a comparison is inappropriate. A fair approach is to compare our ICO model individually with each PIP variant.
> > > > >
> > > > > When evaluated in this way, our ICO model consistently demonstrates a significantly better trade-off between feasibility and optimality. For instance, on the TSPTW100 dataset, compared to PIP with $\lambda = 1.0$, our ICO model reduces the infeasibility rate from 4.33% to 1.33%, while simultaneously improving the optimality gap from 0.25% to 0.14%. Similarly, on TSPTW50, ICO reduces the infeasibility rate from 1.56% to 0.50%, and the optimality gap from 0.16% to 0.07%. Given these consistent improvements across multiple settings, we think it would be inaccurate to describe our improvements as merely "slight".
> > > > >
> > > > > ## Response to "... PIP is not included as a comparison baseline ...", "... should cover problems of varying difficulty ...", and your former comments of "...PIP and PIP-D were not reported or compared separately..."
> > > > >
> > > > > Despite the high computational cost of PIP training, we made every effort to complete the additional evaluations as quickly as possible. The latest results, obtained just an hour ago, are reported in Tables R1–R3. The experiments were conducted on both the **Easy and Medium datasets, using PIP and PIP-D as baselines**. We can observe that our proposed ICO method consistently outperforms both PIP and PIP-D across all tested difficulty levels, further validating its effectiveness.
> > > > >
> > > > > Notably, on the TSPDL/TW Medium datasets, our ICO model achieves both the lowest infeasibility rate and the smallest optimality gap, demonstrating a strong overall trade-off. On the Easy dataset, ICO also **achieves a smaller optimality gap while maintaining a 0% infeasibility rate**. These improvements can be attributed to the adaptive nature of our ICO method: it assigns smaller $\lambda$ values to instances with relatively easier constraints. This allows the model to prioritize optimizing the objective value, thereby achieving better optimality without compromising feasibility. As shown in Table R4, the overall values of the assigned $\lambda$ decrease as the problem difficulty decreases, providing concrete evidence for the effectiveness of our adaptive mechanism.
> > > > >
> > > > > ### Table R1. TSPDL50 Medium
> > > > >
> > > > > | Method | Inf. Rate | Avg. Gap | HV (5%, 5%) | Time |
> > > > > | --- | --- | --- | --- | --- |
> > > > > | PIP | 0.26% | 2.74% | 0.428 | 70s |
> > > > > | PIP-D | 0.23% | 2.98% | 0.385 | 70s |
> > > > > | PIP + ICO | **0.09%** | **2.28%** | **0.534** | 68s |
> > > > >
> > > > > ### Table R2. TSPTW50 Medium
> > > > >
> > > > > | Method | Inf. Rate | Avg. Gap | HV (5%, 5%) | Time |
> > > > > | --- | --- | --- | --- | --- |
> > > > > | PIP | 0.36% | 2.22% | 0.516 | 91s |
> > > > > | PIP-D | 0.23% | 2.62% | 0.454 | 91s |
> > > > > | PIP + ICO | **0.16%** | **1.66%** | **0.647** | 90s |
> > > > >
> > > > > ### Table R3. TSPTW50 Easy
> > > > >
> > > > > | Method | Inf. Rate | Avg. Gap | HV (5%, 5%) | Time |
> > > > > | --- | --- | --- | --- | --- |
> > > > > | PIP | 0.00% | 2.10% | 0.580 | 92s |
> > > > > | PIP-D | 0.00% | 1.95% | 0.610 | 92s |
> > > > > | PIP + ICO | 0.00% | **1.69%** | **0.662** | 90s |
> > > > >
> > > > > ### Table R4. Statistics of $\lambda$ values across different difficulty levels
> > > > >
> > > > > | Difficulty | Median | Upper quarter percentile | Lower quarter percentile |
> > > > > | --- | --- | --- | --- |
> > > > > | Easy | 0.10 | 0.12 | 0.10 |
> > > > > | Medium | 0.10 | 0.32 | 0.10 |
> > > > > | Hard | 0.29 | 1.18 | 0.10 |

---

> > > > > ### Author Response · Authors · 2025-08-09
> > > > > **(2/2) Response**
> > > > >
> > > > > Moreover, during these extended experiments, we also observed that **ICO exhibits notably better generalization performance across varying difficulty levels**—a critical property for real-world deployment, where the true constraint hardness of instances is typically unknown in advance. To demonstrate this advantage, we conducted a generalization experiment where models trained solely on **TSPDL-Hard** were directly evaluated on **TSPDL-Mixed**, which contains both **Medium** and **Hard** instances.
> > > > >
> > > > > As shown below, ICO maintains strong performance and consistently outperforms the baselines in this challenging setting, further highlighting the robustness and superiority of our approach. This improved generalization can also be attributed to the adaptive, instance-level formulation of ICO, which enables the model to dynamically adjust to the specific constraint hardness of each instance. Intuitively, **the more diverse the instances are, the more opportunity ICO has to leverage its flexibility**, resulting in a more robust and effective approach.
> > > > >
> > > > > ### Table R5. Generalization to TSPDL50 mixed
> > > > >
> > > > > | Method | Inf. Rate | Avg. Gap | HV (10%, 10%) | Time |
> > > > > | --- | --- | --- | --- | --- |
> > > > > | PIP | 1.36% | 3.07% | 0.599 | 73s |
> > > > > | PIP-D | 0.17% | 3.83% | 0.607 | 73s |
> > > > > | PIP + ICO | **0.05%** | **2.87%** | **0.709** | 70s |
> > > > >
> > > > > ### Table R6. Generalization to TSPDL100 mixed
> > > > >
> > > > > | Method | Inf. Rate | Avg. Gap | HV (10%, 20%) | Time |
> > > > > | --- | --- | --- | --- | --- |
> > > > > | PIP | 9.56% | 17.09% | 0.006 | 5m |
> > > > > | PIP-D | 3.32% | 17.89% | 0.070 | 5m |
> > > > > | PIP + ICO | **0.43%** | **16.62%** | **0.162** | 5m |
> > > > >
> > > > > ---
> > > > >
> > > > > In summary, based on both the original and newly added experiments, we would like to re-emphasize the key contributions and advantages of our work:
> > > > >
> > > > > 1. **Novel Insight into Lagrangian Training**: We are the first to reveal the limitations of using a policy-level Lagrangian multiplier (i.e., a single $\lambda$) in neural combinatorial optimization. This insight can help to rectify a common but suboptimal practice in the area and encourage a more principled use of Lagrangian training methods, beyond fixed penalty weights.
> > > > >
> > > > > 2. **Instance-Level Multiplier Formulation for Better Trade-Off Performance**: We propose a solid instance-level Lagrangian multiplier formulation, coupled with a multiplier-conditioned policy. Furthermore, we successfully integrated the iterative Lagrangian optimization algorithm into both the training and inference stages. This novel approach enables instance-level adaptive tuning of the multiplier values, which significantly enhances the trade-off between feasibility and optimality-a core research objective in this domain. Furthermore, extensive experiments across Easy, Medium, and Hard datasets, along with comprehensive ablation studies, provide strong empirical evidence for the effectiveness of our approach.
> > > > >
> > > > > 3. **Strong Generalization Across Varying Constraint Difficulties**: Our instance-level formulation exhibits enhanced expressiveness and generalization when handling heterogeneous levels of constraint tightness or difficulty. As noted in recent work [1], most neural solvers tend to overfit to a fixed level of constraint hardness, limiting their applicability in diverse settings. In contrast, our adaptive approach offers a promising solution to this challenge, showing strong potential for real-world deployment where the difficulty level is not known in advance.
> > > > >
> > > > > 4. **Model-Agnostic and Scalable Framework**: The proposed ICO is a scalable, model-agnostic framework that can be seamlessly integrated into various reinforcement learning backbones. Meanwhile, ICO retains efficiency while scaling to large problem sizes, making it a practical and versatile approach.
> > > > >
> > > > > We ensure that all newly added experiments and analyses will be thoroughly incorporated into the revised version of our paper. Thank you once again for taking the time to review our work. We sincerely hope that our response can address your remaining concerns.
> > > > >
> > > > >
> > > > > [1] Rethinking Neural Combinatorial Optimization for Vehicle Routing Problems with Different Constraint Tightness Degrees. In arXiv 2025.

---

> > > > > > ### Comment · Reviewer_JxWh · 2025-08-09
> > > > > > **Reply to the rebuttal**
> > > > > >
> > > > > > Thanks for the diligent response. The presented experimental results have addressed some of my earlier concerns. However, I remain unconvinced about the decision not to follow the established practice (e.g., PIP and MUSLA) of reporting results across multiple difficulty levels, opting instead for a single difficulty level. Given the short rebuttal period, such preparation could have been done in advance. I will raise my overall score, but with a lower confidence level.

---

> > > > > > > ### Author Response · Authors · 2025-08-09
> > > > > > >
> > > > > > > Thank you for your positive feedback! We're glad to hear that your concerns have been addressed. We previously believed that the Hard datasets were sufficiently representative and challenging for comparing state-of-the-art neural solvers. Now, according to your valuable suggestions, we will make sure to report the results of all the difficulty levels for a more comprehensive comparison. Once again, thank you for your insightful feedback!

---

### Official Review · Reviewer_dvDR · 2025-07-02

**Clarity:** 4
**Significance:** 2
**Originality:** 3
**Rating:** 4
**Confidence:** 4

**Summary:**

This paper considers the problem of solving Vehicle Routing Problems (VRPs) with complex constraints, such as time windows, using reinforcement learning (RL). A common approach for solving such constrained optimization problems is the Lagrange multiplier method, which can be combined with RL by introducing a constraint-violation term into the rewards with a multiplier $\lambda$. This paper focuses on the problem of determining the value of the multiplier $\lambda$ for each instance during training, and learning policies that can be conditioned on instance-specific $\lambda$ values. The proposed approach includes pre-training with random $\lambda$ and fine-tuning with adaptive $\lambda$ values, and it uses an extension of POMO [41] as the RL policy. The paper presents numerical results showing improvement over existing approaches in terms of solution quality and constraint satisfaction.

**Questions:**

Is there a reason for using REINFORCE instead of more advanced algorithms, such as PPO?

**Ethical Concerns:**

["NO or VERY MINOR ethics concerns only"]

**Final Justification:**

Overall, this would be a nice addition to the literature on VRPs. The paper seems sound, and it presented well. Additional results, such as the prediction of $\lambda$ values, could make the paper stronger.

**Limitations:**

Yes.

**Paper Formatting Concerns:**

Figure 1 is impossible to read without colors.

**Quality:**

3

**Strengths And Weaknesses:**

Strengths:
* The paper considers an interesting problem with practical applications.
* The proposed approach is described clearly. Generally, the paper is well written and the formalisms are sound.
* The paper provides sufficient details for reproducibility, and the source code for the experiments is available.
* The numerical results are promising, showing improvement over prior work.
* The numerical results also include ablation studies.

Weaknesses:
* The adaptive update of the multiplier $\lambda$ in the fine-tuning stage can only increase $\lambda$ but never decrease it. This seems suboptimal since policy updates may necessitate reduction to $\lambda$. There are experimental results in the appendix on a more advanced, PID-based update of $\lambda$, but these experiments apply the PID-based update only during inference. It is not clear why this approach was not tried during training since it seems like a very intuitive and obvious idea.
* The proposed approach still requires adaptive tuning of $\lambda$ for each problem instance, both during training and inference. The discussion of future work mentions the idea of training a predictor that can quickly predict a (quasi-)optimal $\lambda$ value for a problem instance. This again seems like an obvious idea that should be tried. Including such a predictor (and perhaps training it using an RL paradigm) would make the paper much stronger.
* It is not clear what the "number of timeout nodes" is and why it is included in the penalty term.
* The paper could provide more guidance on how the random distribution of $\lambda$ values should be selected for the pre-training stage.
* "...these two problems pose greater challenges in satisfying constraints compared to CVRPTW and CVRPDL, as the constraint violations of the latter can be addressed more easily by assigning additional vehicles to the violated nodes"
CVRPTW and CVRPDL may have limited number of vehicles, in which case the constraint violations cannot be addressed so easily.
* The copyright file of the source code seems to de-anonymize the paper.

---

> ### Author Rebuttal · Authors · 2025-07-31
>
> Thanks for dedicating your time to reviewing our paper! Your suggestions are very constructive for us to further improve the paper. Please find our detailed response below.
>
> ## Response to “… the update of $\lambda$ can only increase $\lambda$ …”
>
> Thank you very much for your insightful comment. We truly appreciate your suggestion, which has prompted us to further refine our method.
>
> As you pointed out, our original implementation employs **subgradient ascent (SGA)** to update the Lagrange multiplier $\lambda$ during the training phase. While this approach is relatively straightforward, it is also widely adopted in constrained optimization literature and has theoretical convergence guarantees under appropriate learning rate settings. In our experiments on TSPTW and TSPDL, SGA with a carefully tuned learning rate has been effective in guiding $\lambda$ towards values that yield feasible solutions. In our early-stage attempts, we compared Proportional-Integral-Derivative (PID) control with SGA at inference using the same pre-trained policy (trained without iterative $\lambda$ updates). We observed that PID did not consistently outperform SGA in inference. Therefore, we chose the more straightforward SGA strategy as the final implementation.
>
> However, we acknowledge your concern regarding the non-decreasing nature of SGA, which may indeed lead to suboptimal results, especially in combinatorial settings where a fixed learning rate may cause the optima to be skipped. Motivated by your feedback, we have conducted additional experiments to explore more flexible update rules, integrating them not only into inference but also into the training phase.
>
> Specifically, we have explored the following variants:
>
> 1. **Full PID control:** We incorporated PID control into the training phase.
> 2. **ID control (without the proportional term):** Considering the non-smooth nature of combinatorial problems, the proportional term in PID may be over-aggressive. Hence, we removed it and adopted an **Integral-Derivative (ID)** control method.
>
> Below are the experimental results for **TSPTW50-hard** and **TSPDL50-hard**, where the normalized HyperVolume (HV) is computed based on the reference point (5%, 5%):
>
> ### TSPTW50-hard
>
> | | **Infeasibility Rate ↓** | **Average Gap ↓** | **HV ↑** | **Time** |
> | --- | --- | --- | --- | --- |
> | SGA | 0.50% | 0.07% | 0.890 | 90s |
> | PID control | 0.36% | 0.10% | 0.910 | 90s |
> | ID control | 0.19% | 0.09% | **0.940** | 90s |
>
> ### TSPDL50-hard
>
> |  | **Infeasibility Rate ↓** | **Average Gap ↓** | **HV ↑** | **Time** |
> | --- | --- | --- | --- | --- |
> | SGA | 0.01% | 2.32% | 0.535 | 68s |
> | PID control | 0.12% | 2.23% | 0.541 | 68s |
> | ID control | 0.05% | 2.23% | **0.548** | 68s |
>
> The experimental results show that the **PID** and **ID** controls consistently offer improvements over the original SGA method. In particular, ID control achieves the highest HV scores on both benchmarks. We will revise the paper to include these findings. Thank you again for your valuable feedback.
>
> ## Response to “... still requires adaptive tuning of $\lambda$ for each problem instance...”
>
> Thank you very much for this insightful comment. First, we would like to emphasize that although our proposed method involves instance-specific tuning of the multiplier $\lambda$, the increased overhead during training and inference is relatively small. As demonstrated in Table 2, our method can achieve superior performance to the PIP baseline with only two iterations at the inference stage, where the additional overhead is affordable. During training, we control the number of forward and backward passes to match that of the baseline methods by reducing the training instances at the fine-tuning stage, ensuring that the overall training cost remains similar.
>
> We fully agree that training a predictor to estimate a near-optimal $\lambda$ for each instance is a promising direction. In response to your valuable suggestions, we have explored this idea by training a network to distinguish the hard instances requiring high $\lambda$ values. This network makes classification based on instance-level features, i.e., all node features. Since most instances can be directly solved by our ICO policy without iterations, the positive and negative samples are extremely imbalanced. To address this issue, the focal loss [1] is applied to adaptively assign the weights of each sample.
>
> We collect 100k instances and labels of TSPTW50 to train the classifier for 50 epochs. However, the resulting prediction error was prohibitively high: the classifier only has 12% recall in identifying the hard instances, which is far from deployment. We hypothesize that this is due to the limited expressiveness of instance features. Note that the instances are drawn from a single distribution and possess similar constraint characteristics, making them difficult to distinguish in terms of constraint hardness.
>
> Nevertheless, your comment has inspired us to consider a new method beyond static prediction. Specifically, since the rollout process in our model is sequential rather than one-shot, it is possible to dynamically adapt the value of $\lambda$ during the solution construction process. We are currently investigating a strategy that leverages partial solution information—such as constraint tightness and action entropy observed in earlier decision steps—to guide the adjustment of $\lambda$ in later steps. This approach could enable a more effective initialization of $\lambda$ in the first iteration, and has the possibility to be integrated into the RL paradigm, as you suggested.
>
> We sincerely apologize for not being able to complete this experiment within the limited timeframe of the current author response phase. We greatly appreciate your insightful suggestion, and will make our best effort to include the corresponding experimental results in the next discussion phase.
>
> [1] Focal Loss for Dense Object Detection. TPAMI 2020.
>
> ## Response to “... the number of timeout nodes...”
>
> Thank you for your thoughtful comment. In the methodology section of our paper, we assume TSPTW as the example problem to illustrate our approach. In this context, “timeout nodes” refer to those violating time window constraints. We sincerely apologize for its ambiguity and will revise to clarify this definition. Regarding the inclusion of this term in the reward, this design choice is based on empirical findings from prior work (i.e., PIP), where minimizing this heuristic penalty enhances constraint satisfaction. To ensure a fair and consistent comparison, we follow the same reward structure in our implementation. We will make the necessary revisions to explain this aspect.
>
> ## Response to “... more guidance on how the random distribution of $\lambda$ values should be selected...”
>
> Thank you for your thoughtful comment. Our guiding principle in selecting the distribution of $\lambda$ is to align it with the empirical frequency of constraint violations. In our experiments, we observed that the trained policy often leads to small violations, resulting in a long-tailed distribution (see Appendix F.5). Thus, we adopted a triangular distribution that emphasizes smaller $\lambda$ values, better matching the empirical structure. This principle is potentially generalizable, as such statistics are easy to obtain in most cases. We hope this explanation addresses your concern. Thank you again for your valuable feedback.
>
> ## Response to “... CVRPTW and CVRPDL may have limited number of vehicles...”
>
> Thank you for your insightful comment. We agree that in many applications, CVRPTW and CVRPDL often involve a limited number of vehicles, where constraint violations cannot be resolved by simply adding vehicles. Our original statement was based on the common experimental setting, assuming unlimited vehicles, and our intention is to highlight the relative difficulty of constraints in TSPTW and TSPDL. We will revise the paper to rectify this statement.
>
> Besides, we also evaluate CVRPTW under a limited fleet size in Appendix F.1. Our ICO method still outperforms baselines in this harder setting, further demonstrating its effectiveness and generality.
>
> ## Response to “Is there a reason for using REINFORCE instead of more advanced algorithms, such as PPO?”
>
> Thank you very much for your insightful question. To the best of our knowledge, earlier works in neural combinatorial optimization [1, 2] have indeed explored actor-critic algorithms such as PPO, following the prevailing practices within the RL community. However, a subsequent seminal study [3] reported that, somewhat counterintuitively, these actor-critic methods tend to underperform compared to the classical REINFORCE algorithm in this specific context (see Table 1 and Figure 3 in [3]).
>
> However, the mechanisms behind these observations remain an open, theoretically unvalidated question. One plausible explanation for this phenomenon is the inherent difficulty of training a reliable critic in high-dimensional combinatorial state-action spaces. The complexity of the solution space may hinder the critic's ability to provide accurate value estimates. In contrast, REINFORCE—being a critic-free method—relies directly on the observed objective values to compute policy gradients. This can allow for more accurate advantage estimation, thereby leading to superior performance in combinatorial optimization tasks.
>
> We hope this clarifies the rationale behind the choice of REINFORCE in this setting and truly appreciate your question and the opportunity to discuss this further.
>
> [1] Neural Combinatorial Optimization with Reinforcement Learning. ICLR 2017.
>
> [2] Reinforcement Learning for Solving the Vehicle Routing Problem. NeurIPS 2018.
>
> [3] Attention, Learn to Solve Routing Problems! ICLR 2019.
>
> ---
>
> **We hope that our response has addressed your concerns, but if we missed anything, please let us know.**

---

> > ### Comment · Reviewer_dvDR · 2025-08-06
> >
> > Thank you for the detailed response and the additional experimental results! They have answered my questions.
> >
> > However, I do not agree with the interpretation of the results of Ref. [3] (Attention, Learn to Solve Routing Problems! ICLR 2019). If I am not mistaken, this prior works considers a critic for the baseline value $b(s)$, which is different from using a critic for the return (as is done by PPO using GAE). Further, this prior seems to consider only REINFORCE-style policy gradient, not more recent methods, such as PPO or TRPO, that take multiple gradient steps. So I do not agree with interpreting Ref. [3] as a conclusive proof that REINFORCE works best.
> >
> > Overall, I believe that the paper makes a valuable (but somewhat limited) contribution in its current form. If it were expended with additional results on applying (P)ID during training and studying the prediction of $\lambda$ values, I would recommend acceptance more strongly.

---

> > > ### Author Response · Authors · 2025-08-06
> > >
> > > Thank you for taking the time to engage in further discussion. We are pleased to hear that our rebuttal has addressed your previous questions.
> > >
> > > ## Revision to include (P)ID control into training and study the prediction of $\lambda$ values
> > >
> > > Thank you once again for your valuable and constructive suggestions on these two aspects, which have been instrumental in improving our work. Below, we describe the specific efforts we have made in response, as well as how we plan to revise the paper accordingly:
> > >
> > > 1. **Integrating (P)ID control into the training phase.** We have conducted additional experiments to incorporate (P)ID control into the training process. The results, as presented in our initial rebuttal, indicate that training with (P)ID control leads to improved performance on both the TSPTW and TSPDL datasets. In the revised paper, we will update Section 3—particularly Subsections 3.1 and 3.2—to reflect the integration of (P)ID control in both the inference and fine-tuning stages. Additionally, Section 4 (Experiments) will be revised to include the new results. Specifically, we will report the performance of the ICO (PID) variant alongside the existing ICO (SGA) method in the main results table.
> > >
> > > 2. **Studying the prediction of $\lambda$ values.** We appreciate this insightful suggestion. Based on our new experiments, we found that predicting the hardness of instances purely from instance-level features is highly challenging, as constraint-related features are generated in a very similar manner across instances. Instead, we propose a more adaptive approach that adjusts $\lambda$ values dynamically during the solution construction process. This strategy takes into account partial solution information—such as current constraint tightness—to identify potentially infeasible instances and increase their corresponding $\lambda$ values accordingly. We refer to this method as **one-iteration adjustment** for $\lambda$, which is inherently an adjustment strategy, but it completes in a single iteration to enhance efficiency. As promised in our last response, we will report the experimental results of this method in the new discussion (We cannot finish the experiment due to time limit in the rebuttal phase). Experimental results on TSPTW50 and TSPTW100 (shown below) demonstrate that this strategy effectively improves performance with minimal additional computational cost. We will incorporate this inference strategy into both the methodology and experiments sections of the revised paper.
> > >
> > > ### TSPTW50
> > >
> > > | Method | Inf. rate | Avg. Gap | HV | Time |
> > > | --- | --- | --- | --- | --- |
> > > | ICO (T=1) | 2.14% | 0.10% | 0.959 | 9s |
> > > | ICO (One-iteration adjustment) | **1.99%** | 0.10% | **0.961** | 11s |
> > >
> > > ### TSPTW100
> > >
> > > | Method | Inf. rate | Avg. Gap | HV | Time |
> > > | --- | --- | --- | --- | --- |
> > > | ICO (T=1) | 13.20% | 0.16% | 0.840 | 41s |
> > > | ICO (One-iteration adjustment) | **11.76%** | 0.17% | **0.852** | 47s |
> > >
> > > We will make sure to include the improved techniques and the new results in the final version. We sincerely thank you for your valuable feedback again.
> > >
> > > ## Discussions about PPO and REINFORCE
> > >
> > > We agree that the results of [3] primarily compare REINFORCE-based algorithms with different baselines, making the comparison less conclusive. We sincerely apologize for this oversight. After carefully reviewing the literature again, we found that the RL4CO paper [4] has done a comprehensive comparison of RL algorithms, as shown in Table E.1 and Figure E.2 in its Appendix. Briefly, Table E.1 aligns the total number of training instances and shows that AM-PPO does not consistently outperform REINFORCE-based methods such as AM-XL and POMO [5]. Figure E.2, which controls the total number of evaluations (noting that REINFORCE requires more evaluations due to baseline estimation), suggests that PPO offers better sample efficiency. These observations suggest that PPO may be advantageous when evaluation costs are high, but it offers limited benefit in inexpensive evaluation scenarios, such as TSP and CVRP.
> > >
> > > In our work, we just follow the common practice of using REINFORCE to ensure fair comparisons. The opinions and experimental observations presented here are only intended for reference. We believe that the choice between PPO and REINFORCE is still an open, theoretically unvalidated question. We will add these discussions and clearly indicate it as an interesting future work for NCO. Thank you again for your valuable question.
> > >
> > >
> > > [1] Neural Combinatorial Optimization with Reinforcement Learning. ICLR 2017.
> > >
> > > [2] Reinforcement Learning for Solving the Vehicle Routing Problem. NeurIPS 2018.
> > >
> > > [3] Attention, Learn to Solve Routing Problems! ICLR 2019.
> > >
> > > [4] RL4CO: An extensive reinforcement learning for combinatorial optimization benchmark. KDD 2025.
> > >
> > > [5] POMO: Policy Optimization with Multiple Optima for Reinforcement Learning. NeurIPS 2020.

---

> > > > ### Comment · Reviewer_dvDR · 2025-08-06
> > > >
> > > > Regarding REINFORCE vs. PPO (or any other off-the-shelf algorithm): I would like to clarify that this was just a suggestion for potentially obtaining stronger results. Since both algorithms are off-the-shelf and applicable in a straightforward manner, the choice between them is orthogonal to the contributions of this paper, so I did not consider it to be a weakness in my review.

---

> > > > > ### Author Response · Authors · 2025-08-07
> > > > >
> > > > > Thank you very much for acknowledging our contributions and for providing valuable suggestions to improve our work. We fully agree that the choice between PPO and REINFORCE represents an orthogonal direction for further enhancing performance. We sincerely appreciate you pointing this out, as it has inspired meaningful discussions and reflections on the selection of reinforcement learning algorithms, which may lead to further improvements in future work.
> > > > >
> > > > > Following your previous suggestions, we ensure that the newly introduced techniques—such as the PID-control-based training and the efficient prediction and adjustment of $\lambda$ values—will be clearly and appropriately incorporated into our revised paper, as described in our earlier response.
> > > > >
> > > > > Thank you again for your insightful comments and constructive discussions.

---

### Comment · Area_Chair_ExqZ · 2025-08-06

Dear Reviewers,

As the deadline for the author-reviewer discussion approaches, we kindly ask that you review the rebuttal at your earliest convenience. At present, two of you have not yet engaged in the discussion—we would appreciate it if you could actively participate in the dialogue with the authors.

Please note that simply clicking the acknowledgement button without substantive input is not sufficient, as this may result in a flag for insufficient review. Additionally, delaying your response until the last moment will also trigger a flag.

Thank you for your understanding and for dedicating your valuable time to the review process.

---

### Note · Authors · 2025-08-12

Dear AC and Reviewers,

We sincerely thank you for your time, constructive feedback, and insightful discussions. We appreciate your valuable suggestions, which have indeed helped improve our work. We are very glad that your concerns have been addressed during the discussion phase, and now you are all positive about our work. Here, we provide a brief summary of each reviewer’s key points and our responses to facilitate the final decision.

1. **Reviewer dvDR** provided two suggestions to further improve the techniques within our proposed ICO framework: integrating PID control into training and enabling faster prediction/adjustment of the multiplier $\lambda$. In response, we conducted new experiments of incorporating (P)ID control, and proposed a one-iteration adjustment strategy to improve efficiency (also validated by new experiments). These efforts led to productive and insightful interactions with the reviewer.

2. **Reviewer JxWh** questioned the generality of our framework and its performance on easier datasets. To address this, we combined ICO with a new backbone, RELD, demonstrating its versatility. We also added experiments on Easy, Medium, and even Mixed-hardness datasets using both PIP and PIP-D baselines. The reviewer acknowledged these improvements and raised his/her score accordingly.

3. **Reviewer D8c6** raised concerns regarding the novelty and empirical impact of our method. We clarified our contributions from both conceptual and technical perspectives, highlighted overall performance gains, and emphasized the model-agnostic and scalable nature of our approach. The reviewer found these clarifications satisfactory and increased the overall score.

4. **Reviewer 5viD** focused on the formulation of HyperVolume and the performance on large-scale and real-world datasets. We provided a detailed explanation of the standard HyperVolume formulation and added new experiments on larger-scale (e.g., TSPTW200) and real-world instances (e.g., TSPLIB). Though the reviewer did not respond, we believe these responses addressed the concerns. We hope the reviewer will consider his/her willingness to increase the score in the initial comment.

We make sure that all additional experiments, techniques, and discussions will be clearly reflected in the revised version. Thank you very much again for your valuable feedback and support.

Best regards,

The authors

---

### Decision · Program_Chairs · 2025-09-17

**Decision:**

Reject

**Comment:**

The authors propose an instance-level adaptive constrained optimization framework that reformulates the Lagrangian dual problem by assigning each instance its own multiplier. This framework enables the solution of vehicle routing problems with complex constraints, such as time windows and draft limits. Its main strengths lie in the sound motivation, clear writing, and solid technical contribution. The work has the potential to advance research on handling complex constraints in vehicle routing problems. In the rebuttal, the authors clarified key concepts and experimental settings, and presented additional results on training efficiency and RELD.

While all reviewers gave positive scores, their final remarks were generally more neutral. Concerns remain regarding the completeness of experiments across different difficulty levels. The authors reported better results with the suggested simple PID and ID methods, which should be investigated more thoroughly and integrated into the main paper. Another issue is the significance of the work, which appears limited given the results in Table 2. Incorporating PID and ID is necessary to achieve results that clearly surpass the baselines. Finally, questions about generalization persist: although partially addressed with additional tests on TSPLIB, the TSPTW benchmark would be more appropriate in this context.